# Mbnl1 and Mbnl2 regulate brain structural integrity in mice

Naomi S. Sta Maria[1,4], Chenyu Zhou[2,4], Se Jung Lee[2,4], Parvin Valiulahi[2], Xiandu Li[3], Jongkyu Choi[2], Xiaodan Liu[1], Russell Jacobs[1], Lucio Comai [ID][3✉] & Sita Reddy [ID][2✉]

Myotonic Dystrophy Type I (DM1) patients demonstrate widespread and variable brain structural alterations whose etiology is unclear. We demonstrate that inactivation of the Muscleblind-like proteins, Mbnl1 and Mbnl2, initiates brain structural defects. 2D FSE T2w MRIs on 4-month-old $Mbnl1^{+/-}/Mbnl2^{-/-}$ mice demonstrate whole-brain volume reductions, ventriculomegaly and regional gray and white matter volume reductions. Comparative MRIs on 2-month-old $Mbnl1^{-/-}$, $Mbnl2^{-/-}$ and $Mbnl1^{-/-}/Mbnl2^{+/-}$ brains show genotype-specific reductions in white and gray matter volumes. In both cohorts, white matter volume reductions predominate, with Mbnl2 loss leading to more widespread alterations than Mbnl1 loss. Hippocampal volumes are susceptible to changes in either Mbnl1 or Mbnl2 levels, where both single gene and dual depletions result in comparable volume losses. In contrast, the cortex, inter/midbrain, cerebellum and hindbrain regions show both gene and dose-specific volume decreases. Our results provide a molecular explanation for phenotype intensification in congenital DM1 and the variability in the brain structural alterations reported in DM1.

[1] Department of Physiology and Neuroscience, Keck School of Medicine, University of Southern California, Los Angeles, CA, USA. [2] Department of Biochemistry and Molecular Medicine, Keck School of Medicine, University of Southern California, Los Angeles, CA, USA. [3] Department of Molecular Microbiology and Immunology, Keck School of Medicine, University of Southern California, Los Angeles, CA, USA. [4] These authors contributed equally: Naomi S. Sta Maria, Chenyu Zhou, and Se Jung Lee. ✉email: comai@usc.edu; sitaredd@usc.edu

Myotonic dystrophy type I (DM1) is an RNA-dominant multi-system disorder in which the brain is prominently affected[1–6]. DM1 pathology results primarily from the expression of toxic expanded CUG repeat RNA which dysregulates RNA biology[7]. RNA processing, localization, and translation defects arise from CUG-expansion RNA-mediated functional alterations of RNA-binding proteins, including the muscleblind-like (MBNL) proteins, CUG-BP1 and hnRNPH[8–15]. In addition, CUG-expansion RNAs localize the SHARP transcription factor from the nucleus to the cytoplasm in patient myoblasts to alter steady-state mRNA levels[16] and repeat-associated non-AUG (RAN) translation can result in the generation of toxic peptides[17]. SHARP/Mint inactivation causes severe postnatal brain hypoplasia in mice[18] and RAN-translation peptides have been shown to play an important role in the degenerative brain changes observed in ALS and frontotemporal dementia[19]. The molecular mechanisms underpinning DM1 brain structural alterations are however currently not fully understood.

DM1 is characterized by genetic anticipation and symptom variability[1–4]. Genetic anticipation or an increase in symptoms and severity within a pedigree is a consequence of the progressive expansion of the CTG-repeat tract between generations[1,20]. Additional sources of phenotypic variability are the stochastic CTG-repeat expansions that occur in somatic cells with time[3,21]. Both anticipation and symptom variability are pronounced in the brain, with large, mostly maternally inherited CTG expansions, resulting in severe pathology characterized by mental retardation and autism-like features in the congenital form of the disease[1,2,20]. Smaller expansions inherited from either parent result in milder adult-onset or moderately affected juvenile-onset DM1, where patients can demonstrate mental retardation, deficits in memory and learning, with visuospatial learning and reconstruction being prominently affected, dysexecutive syndrome, and excessive daytime sleepiness. Other behavioral and emotional alterations include anxiety, depression, apathy, attention-deficit hyperactivity disorder, obsessive compulsive disorder, and avoidant personality traits[1,2].

In congenital DM1 patients, magnetic resonance imaging (MRI) studies demonstrate microcephaly, global brain atrophy, and prominent ventricular enlargement[3,4,22,23]. As ventricular enlargement, which is documented in greater than 50% of congenital DM1 infants in some studies, is observed at birth, brain structural defects in this group must in part be developmental in origin[1–4,22]. In adult-onset and juvenile-onset DM1, ventricular enlargement is less prominent but can progress with time[2,23]. All three DM1 patient classes show regional gray matter atrophy in the cortex, the hippocampus, the basal ganglia and other subcortical regions, and in the cerebellum[4,24]. White matter hyperintensities in the external capsule, the cortex, subcortical deep white matter, and in the periventricular regions are observed in all three DM1 classes[3,4,23,24]. The structure of normal appearing white matter can show widespread impairments, demonstrating both reductions in fractional anisotropy and increased mean diffusivity[3]. Such alterations are recorded in the corpus callosum, the capsule, and in a majority of the association, projection, and commissural fibers[3,24]. Decreases in white matter volumes in the corpus callosum, the fornix, the cingulum bundle, and in the subcortical regions of all lobes are observed in DM1 with the thinning of the corpus callosum being particularly prominent in congenital DM1[4,23].

Longitudinal studies, although rare, demonstrate progressive degenerative changes in white matter, cerebral atrophy, and increases in ventricular volume with time[22,25]. The effect of age and disease-duration studies with matched cohorts further supports progressive gray- and white matter atrophy and increases in ventricular volume with time[23,24,26]. Possible correlations between structural impairments and behavioral alterations in DM1 are unresolved with some studies demonstrating correlative changes in behavior and white matter defects[24,27,28]. Thus, neuroimaging in DM1 patients demonstrates widespread brain involvement and lends support for gray matter atrophy and loss of connectivity between gray matter regions occurring secondarily to white matter breakdown contributing to behavioral changes in DM1 patients[2–4,24,28].

As one or more molecular defects may work coordinately to impact brain structural integrity in DM1, deciphering the molecular mechanisms that initiate such changes is largely intractable in patients. We therefore carried out MRI studies in mouse models where Mbnl1 and Mbnl2 are inactivated either singly or in composite. This analysis shows widespread gray- and white matter defects resulting from Mbnl1 and Mbnl2 loss and identifies a regulatory role for the Mbnl1 and the Mbnl2 proteins in maintaining the structural integrity of the brain. As the brain structural alterations that manifest with Mbnl1/2 loss recapitulate several prominent DM1 brain structural defects, this study demonstrates that a key mechanism driving the brain structural alterations in DM1 patients is the functional inactivation of the MBNL1 and the MBNL2 proteins by the CUG-expansion RNAs. Correlative changes in hippocampal and white matter volume reductions with the behavioral alterations reported in mouse models depleted of Mbnl1 and Mbnl2[29,30] support the potential of brain structural defects working in concert with the functional alterations in MBNL1/2-target RNAs in driving behavioral dysfunction in DM1 patients.

## Results

To test the hypothesis that Mbnl1 and Mbnl2 regulate the structural integrity of the brain, we developed 4-month-old female (4mo) 129sv $Mbnl1^{+/+}/Mbnl2^{+/+}$ ($n = 5$) and 129sv $Mbnl1^{+/-}/Mbnl2^{-/-}$ ($n = 5$) cohorts as described in the "Methods". As the $Mbnl1^{-/-}/Mbnl2^{-/-}$ genotype demonstrates an embryonic lethal phenotype and because Mbnl2 levels are higher in the brain when compared with Mbnl1 levels[9], $Mbnl1^{+/-}/Mbnl2^{-/-}$ mice should afford the largest Mbnl1/2 reductions in the brain. Four-month-old cohorts were chosen because T2 weighted and diffusion-weighted MRI in mice have been shown that the most dramatic postnatal changes in cortex thickness and myelination are largely over by three months[31].

$Mbnl1^{+/-}/Mbnl2^{-/-}$ mice demonstrate widespread brain structural alterations. $Mbnl1^{+/+}/Mbnl2^{+/+}$ and $Mbnl1^{+/-}/Mbnl2^{-/-}$ cohorts were subjected to 2D FSE T2w MRI and a montage of extracted whole-brain slices for each subject was assembled with or without overlaid manually drawn, color-filled regions of interest for qualitative assessments. Volumetric measurements of the ROIs were extracted for quantitative analysis as described in the Methods. These montages demonstrated that the total number of slices were reduced by 1–4 slices with regional differences being observed in $Mbnl1^{+/-}/Mbnl2^{-/-}$ brains when compared with $Mbnl1^{+/+}/Mbnl2^{+/+}$ brains (Fig. 1, Supplementary Figs. 3 and 4). Consistent with this observation, whole-brain volumes were reduced ~9% in $Mbnl1^{+/-}/Mbnl2^{-/-}$ mice when compared with age-matched controls ($p = 0.0001$) (Fig. 2a, b, Table 1). Concurrent with reduced whole-brain volumes, mean apparent ventricle volume was ~126% increased ($p = 0.0002$) and the apparent ventricle volume to the whole-brain volume ratio was ~152% increased ($p = 0.0001$) in $Mbnl1^{+/-}/Mbnl2^{-/-}$ mice when compared with $Mbnl1^{+/+}/Mbnl2^{+/+}$ controls (Fig. 2c–e, Table 1).

To define the contributions of region-specific volume alterations to the observed shrinkage in whole-brain volumes, all identifiable

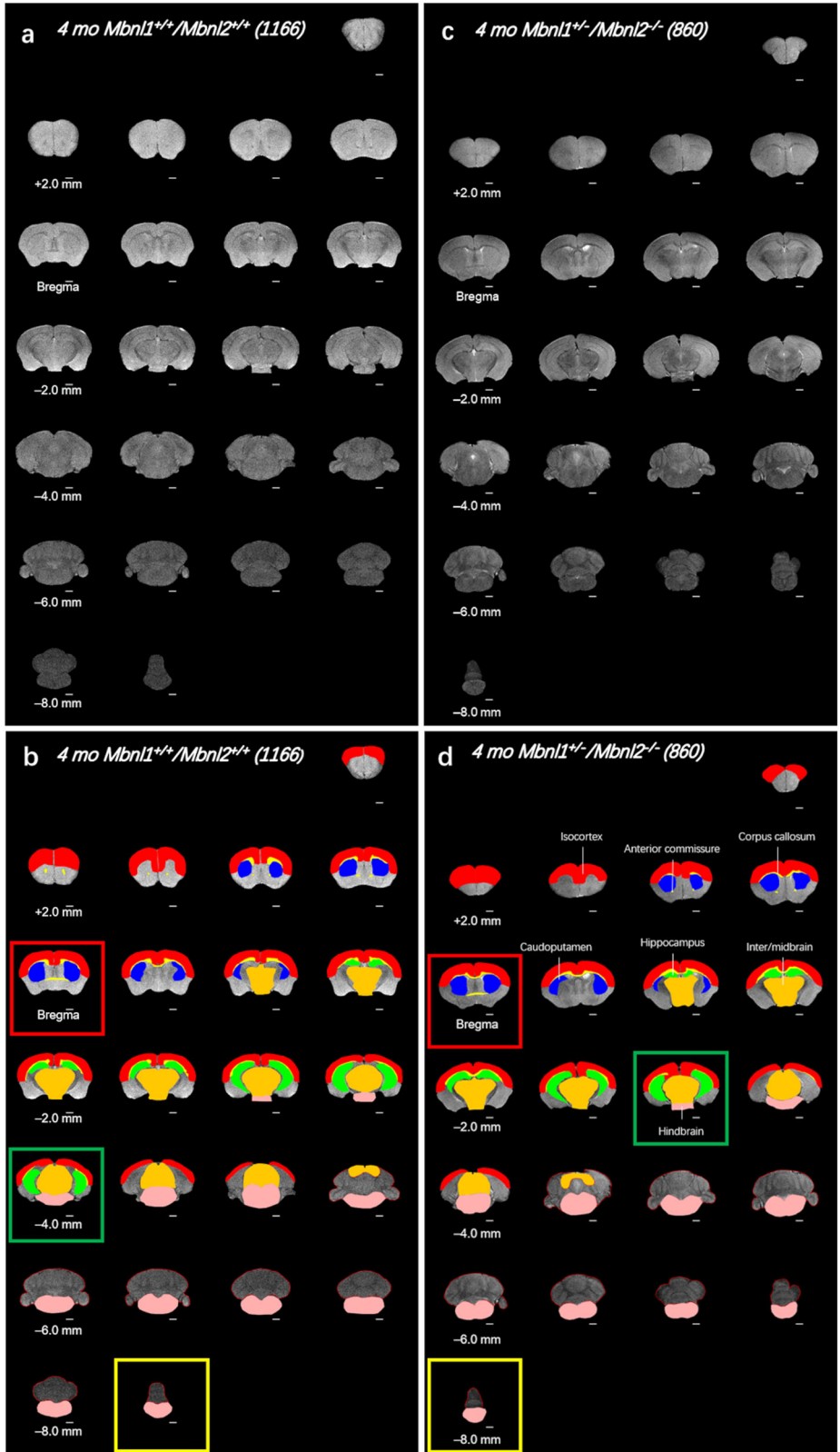

white- and gray matter regions of interest were measured in both cohorts. For the white matter, the anterior commissure and the corpus callosum were studied. The defined corpus callosum region in this study included the external capsule white matter areas. Mean corpus callosum/external capsule volumes in the 4-month-old female $Mbnl1^{+/-}/Mbnl2^{-/-}$ mice showed a significant reduction (~21% decrease) in volume ($p = 0.0076$) when compared

with age matched $Mbnl1^{+/+}/Mbnl2^{+/+}$ mice (Fig. 3a, b; Table 1). $Mbnl1^{+/-}/Mbnl2^{-/-}$ mice exhibited larger reductions in the anterior commissure volume (~49% decrease) at four months ($p = 0.0001$) (Fig. 3c, d; Table 1).

Gray matter volumes were measured in the cerebrum, brainstem, and cerebellum. In the cerebrum, the isocortex, hippocampus, and caudoputamen were measured. $Mbnl1^{+/-}/Mbnl2^{-/-}$ mice

**Fig. 1 Whole-brain FSE T2w MRI slices of 4mo *Mbnl1*$^{+/-}$/*Mbnl2*$^{-/-}$ mice show wide-spread regional volume alterations. a**, **c** A montage of 2D FSE T2w MRI brain slices of representative *Mbnl1*$^{+/+}$/*Mbnl2*$^{+/+}$ (wild-type control) and *Mbnl1*$^{+/-}$/*Mbnl2*$^{-/-}$ mice that underwent in vivo scanning are shown, respectively. Mouse identifier numbers are shown in parenthesis. **b**, **d** Manually defined, color-filled regions of interest (ROI) overlaid onto structural FSE T2w MRI slices are shown. All ROIs that are significantly different between the two genotypes are displayed. Color code: white matter regions, which include the anterior commissure and the corpus callosum/external capsule (yellow). Gray matter regions, which include the isocortex (red), the caudoputamen (blue), the hippocampus (green), the inter-/midbrain (orange), and the hindbrain (pink). Calculated areas from the ROIs were multiplied by the image-slice thickness to generate ROI volumes shown in Figs. 2 and 3. The olfactory bulb was excluded in volume measurements. Slice positions are identified based on Bregma locations. The slice where the anterior commissure is connected between hemispheres approximates Bregma, which is defined as 0.0 mm. Going from rostral to caudal slice positions, boxes indicate structural landmarks that include Bregma (red box), the last slice containing hippocampal regions (green box), and the end of the brain (yellow box). Slice thickness = 0.5 mm. Scale bar = 1 mm.

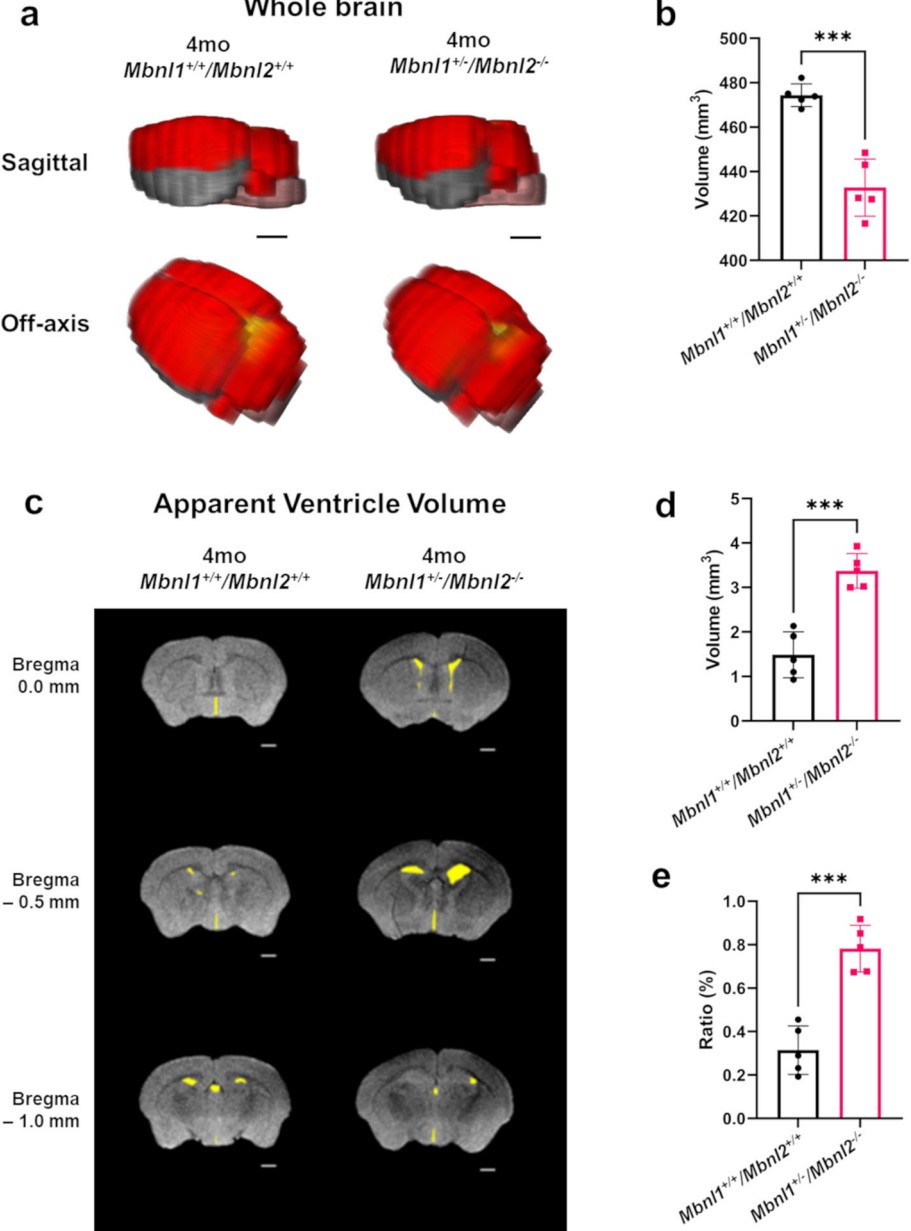

**Fig. 2 Four-month-old *Mbnl1*$^{+/-}$/*Mbnl2*$^{-/-}$ mice exhibit reduced whole-brain volume and enlarged ventricles. a** 3D rendering of representative whole-brain 2D FSE T2w MRIs for 4mo *Mbnl1*$^{+/+}$/*Mbnl2*$^{+/+}$ and *Mbnl1*$^{+/-}$/*Mbnl2*$^{-/-}$ mice. Colored regions are isocortex and cerebellum (red), hindbrain (pink), white matter areas (yellow), and olfactory gray matter areas (gray). The olfactory bulb was excluded in volume measurements. Scale bar = 2.0 mm. **b** Whole-brain measurements (mean ± SD) are shown. **c** Representative axial brain slices from FSE T2w MRI scans where apparent ventricle volumes were defined from hyperintense regions along ventricle locations (yellow filled). Scale bar in 2D axial images denotes 1 mm. **d**, **e** Apparent ventricle volumes and percent apparent ventricle/whole-brain volumes (mean ± SD) are shown. Error bars are standard deviations (SD). In all cases, *$p < 0.05$, **$p < 0.01$, and ***$p < 0.001$.

**Table 1 Summary of brain region volumes (in mm³) from 4-month-old Mbnl1$^{+/+}$/Mbnl2$^{+/+}$ and Mbnl1$^{+/-}$/Mbnl2$^{-/-}$ mice.**

| Region | 4mo Mbnl1$^{+/+}$/ Mbnl2$^{+/+}$ | | 4mo Mbnl1$^{+/-}$/ Mbnl2$^{-/-}$ | | Change (percent) | p-value (Two tailed t-test) |
|---|---|---|---|---|---|---|
| | Mean | SD | Mean | SD | | |
| Anterior commissure | 0.94 | 0.08 | 0.48 | 0.12 | −48.94 | 0.0001 |
| Corpus callosum/external capsule | 8.71 | 0.98 | 6.9 | 0.59 | −20.78 | 0.0076 |
| Caudoputamen | 17.85 | 0.66 | 19.12 | 0.92 | 7.11 | 0.0364 |
| Cerebellum | 75.15 | 5.23 | 73.6 | 6.71 | −2.06 | 0.6935 |
| Isocortex | 104.16 | 1.09 | 95.54 | 3.2 | −8.28 | 0.0005 |
| Isocortex: Bregma+[a] | 35.41 | 1.14 | 35.51 | 1.82 | 0.28 | 0.9198 |
| Isocortex: Bregma-[b] | 68.75 | 1.86 | 60.04 | 2.62 | −12.67 | 0.0003 |
| Hindbrain | 53.97 | 2.71 | 44.44 | 3.85 | −17.66 | 0.0019 |
| Hippocampus | 21.02 | 1.73 | 17.11 | 1.7 | −18.6 | 0.0069 |
| Inter/midbrain | 66.99 | 3.91 | 56.97 | 3.39 | −14.96 | 0.0025 |
| Whole brain | 474.34 | 5.13 | 432.76 | 12.85 | −8.77 | 0.0001 |
| Apparent ventricles[c] | 1.49 | 0.51 | 3.37 | 0.39 | 126.17 | 0.0002 |
| Apparent ventricle/Whole-brain ratio (%) | 0.31 | 0.11 | 0.78 | 0.11 | 151.61 | 0.0001 |
| Number of mice per genotype | 5 | | 5 | | | |

demonstrated significantly reduced isocortex (~8% decrease, $p = 0.0005$) (Fig. 3,e, m; Table 1) and hippocampus volumes (~19% decrease, $p = 0.0069$) (Fig. 3h, n; Table 1) when compared with Mbnl1$^{+/+}$/Mbnl2$^{+/+}$ mice. In the isocortex, volume shrinkage was significant in the posterior Bregma isocortex, defined as Bregma- (~13% decrease, $p = 0.0003$) but not in the anterior Bregma isocortex, defined as Bregma + (Fig. 3f, g; Table 1). In contrast to the isocortex and the hippocampus, the caudoputamen volumes were larger (~7% increase, $p = 0.0364$) (Fig. 3i, o; Table 1) in Mbnl1$^{+/-}$/Mbnl2$^{-/-}$ mice. In the brainstem, both the inter-/midbrain and hindbrain volumes were reduced with the inter-/midbrain region showing an ~15% decrease ($p = 0.0025$) and the hindbrain showing an ~18% decrease ($p = 0.0019$) (Fig. 3j, k, p, q; Table 1) in Mbnl1$^{+/-}$/Mbnl2$^{-/-}$ mice when compared with Mbnl1$^{+/+}$/Mbnl2$^{+/+}$ controls. No significant difference was observed in the mean volumes of the cerebellum in the Mbnl1$^{+/-}$/Mbnl2$^{-/-}$ and the Mbnl1$^{+/+}$/Mbnl2$^{+/+}$ mice (Fig. 3I, Table 1).

Thus, Mbnl1 heterozygosity, in conjunction with Mbnl2 loss, results in global brain atrophy with increased apparent ventricle volumes and widespread white and gray matter volume reductions in the mouse brain at four months of age. The caudoputamen is an exception, demonstrating a volume increase in Mbnl1$^{+/-}$/Mbnl2$^{-/-}$ mice. It is unclear if this change reflects a bona fide increase or is a compensatory alteration in response to other regional gray- and white matter volume shrinkages.

**Mbnl1 and Mbnl2 genotype-specific regional volume reductions are observed in the mouse brain.** As extensive changes in white- and gray matter volumes are observed in 4-month-old female Mbnl1$^{+/-}$/Mbnl2$^{-/-}$ brains, we tested if Mbnl1 and Mbnl2 gene-specific effects are important variables in the maintenance of the structural integrity of the mouse brain. For this analysis, we studied female cohorts of 129sv Mbnl1$^{-/-}$, Mbnl2$^{-/-}$, Mbnl1$^{-/-}$/Mbnl2$^{+/-}$, and control Mbnl1$^{+/+}$/Mbnl2$^{+/+}$ mice at two months of age ($n = 5$/genotype). Two-month-old mice (2mo) were examined for this comparative study as 129sv Mbnl1$^{-/-}$/Mbnl2$^{+/-}$ mice do not generally survive past the age of two months and because a significant attrition of Mbnl1$^{-/-}$ mouse cohorts occurs between the age of 2 and 4 months due to a sudden-death phenotype, presumably from cardiac arrhythmias[32].

The 2-month-old cohorts were subjected to the same 2D FSE T2w MRI protocol and image analysis as the 4-month-old Mbnl1$^{+/-}$/Mbnl2$^{-/-}$ mouse cohorts. The Mbnl1$^{-/-}$/Mbnl2$^{+/-}$ montages demonstrated the most obvious differences, containing in some cases 2–4 fewer slices, when compared with 2-month-old Mbnl1$^{+/+}$/Mbnl2$^{+/+}$ control brains (Fig. 4). Mbnl1$^{-/-}$/Mbnl2$^{+/-}$ whole-brain volume was reduced when compared with Mbnl1$^{+/+}$/Mbnl2$^{+/+}$ brains (~6% decrease, unpaired two-tailed t-test, $p = 0.0179$) (Fig. 5a, b; Table 2). Volumetric region-specific measurements showed similar but not identical changes in Mbnl1$^{-/-}$ and Mbnl2$^{-/-}$ brains, with more widespread and severe changes being observed in most brain regions in the Mbnl1$^{-/-}$/Mbnl2$^{+/-}$ mice (Fig. 4; Supplementary Figs. 5–8; Table 2). Apparent ventricle volume comparisons between the four genotypes were performed using a one-way ANOVA analysis. A significant effect of group was observed in the apparent ventricle volume [(F (3, 16) = 4.782), $p = 0.0145$]. Post hoc testing (Dunnet's multiple test) revealed a significant increase in apparent ventricle volume (~53% increase, $p = 0.014$) and an enlarged apparent ventricle volume to whole-brain volume ratio (~64% increase, $p = 0.0099$) in Mbnl1$^{-/-}$/Mbnl2$^{+/-}$ brains when compared with Mbnl1$^{+/+}$/Mbnl2$^{+/+}$ brains (Fig. 5c, d, e; Table 2). Mbnl1$^{-/-}$ and Mbnl2$^{-/-}$ mice did not demonstrate a change in whole-brain volume or apparent ventricle volume when compared with age-matched Mbnl1$^{+/+}$/Mbnl2$^{+/+}$ controls (Fig. 5, Table 2).

For the white matter, a significant group effect in both the anterior commissure [F (3, 16) = 9.370, $p = 0.0008$] and the corpus callosum/external capsule regions [F (3, 16) = 16.33, $p < 0.0001$] was observed in the 2-month-old cohorts. For the corpus callosum/external capsule, post hoc Dunnett's multiple-comparison test demonstrated comparable shrinkages in the Mbnl1$^{-/-}$ (~29% decrease, $p = 0.001$) and the Mbnl2$^{-/-}$ (~31% decrease, $p = 0.0004$) mice, with the volume decrease being more prominent in the Mbnl1$^{-/-}$/Mbnl2$^{+/-}$ (~43% decrease, $p = 0.0001$) mice when compared with age-matched Mbnl1$^{+/+}$/Mbnl2$^{+/+}$ controls (Fig. 6a, b; Table 2). Post hoc Dunnett's multiple-comparison test showed that the Mbnl2$^{-/-}$ (~31% decrease, $p = 0.0008$) and the Mbnl1$^{-/-}$/Mbnl2$^{+/-}$ (~29% decrease, $p = 0.0015$) mice, but not the Mbnl1$^{-/-}$ mice, demonstrated significantly reduced anterior commissure volumes when compared with 2-month-old Mbnl1$^{+/+}$/Mbnl2$^{+/+}$ control mice (Fig. 6c, d; Table 2).

For the gray matter regions, in the cerebrum, no significant differences were observed between the four genotypes at two

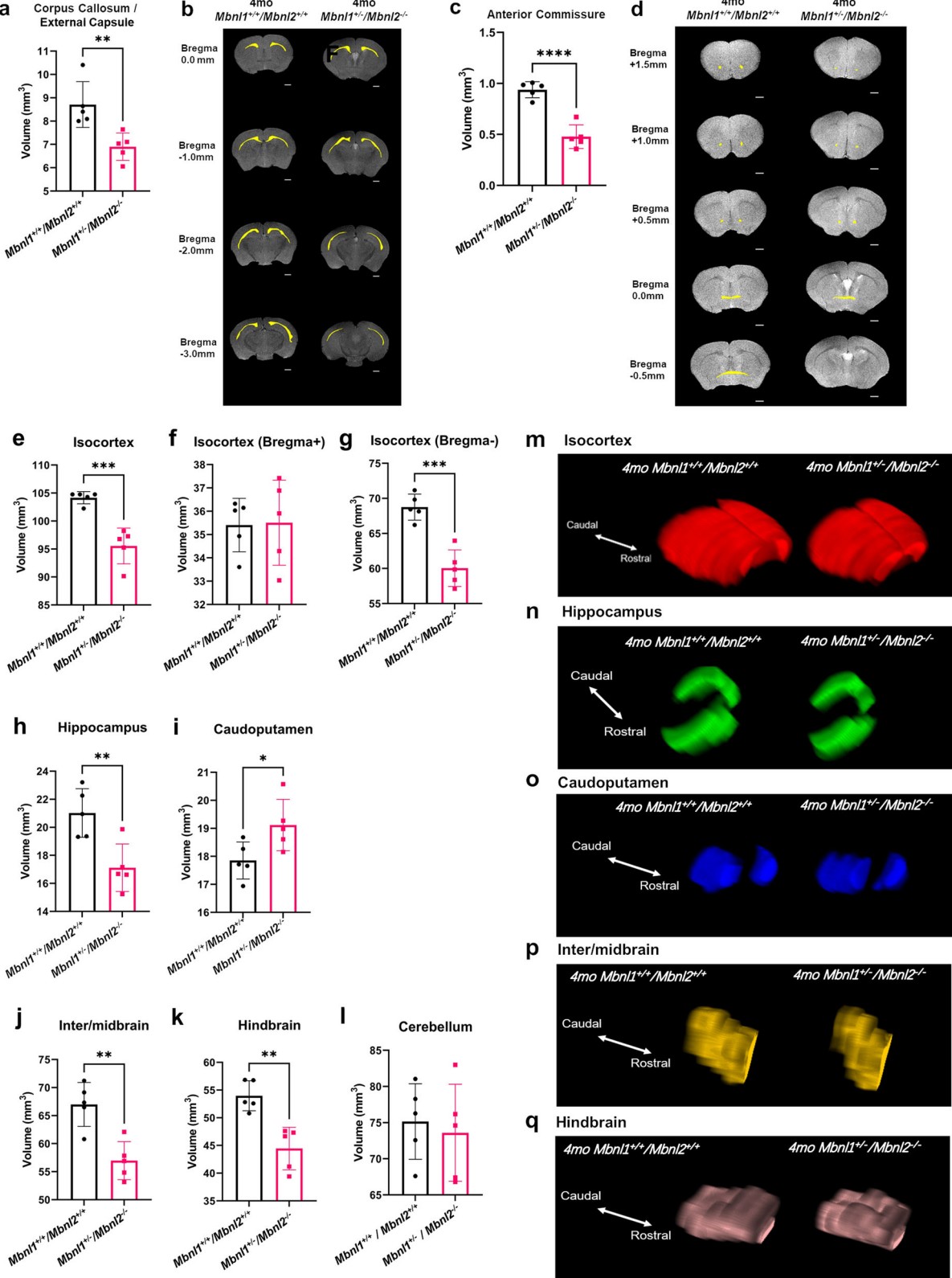

**Fig. 3 White- and gray matter regional volume changes in 4mo *Mbnl1*$^{+/-}$/*Mbnl2*$^{-/-}$ mice. a** Corpus callosum/external capsule measurements (mean ± SD) are shown. **b** Representative axial brain slices from FSE T2w MRI scans with corpus callosum/external capsule volumes overlaid in yellow. **c** Anterior commissure measurements (mean ± SD) are shown. **d** Representative axial brain slices from FSE T2w MRI scans with anterior commissure volumes overlaid in yellow. Scale bar in 2D axial images denotes 1 mm. **e–l** Isocortex, isocortex (Bregma+), isocortex (Bregma−), hippocampus, caudoputamen, inter-/midbrain, hindbrain, and cerebellum volume measurements (mean ± SD) are shown. **m–q** 3D caudal–rostral projections of representative isocortex, hippocampus, caudoputamen, inter-/midbrain, and hindbrain regions are shown. Error bars are standard deviations (SD). In all cases, *$p < 0.05$, **$p < 0.01$, ***$p < 0.001$, and ****$p < 0.0001$.

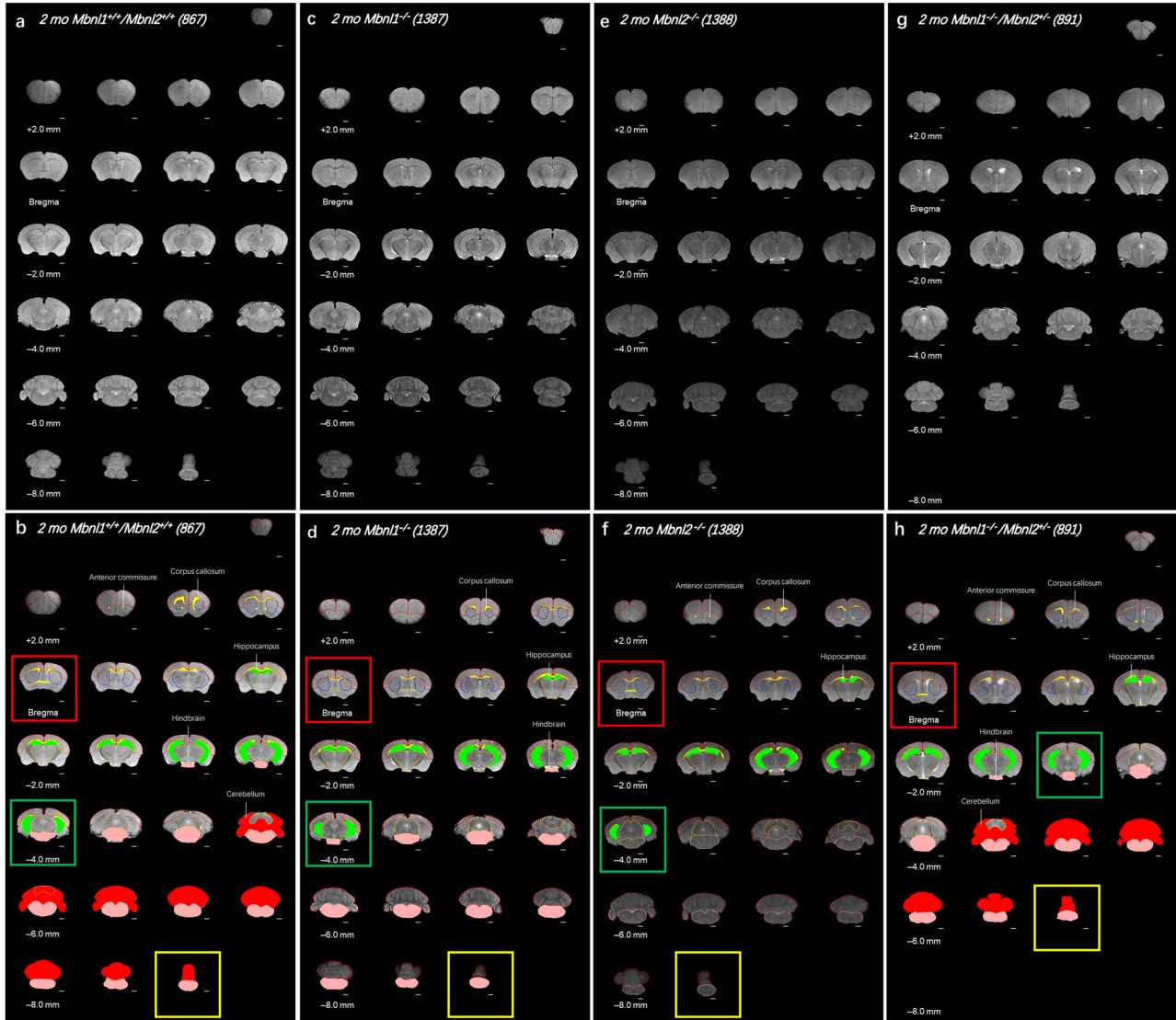

**Fig. 4 Whole-brain FSE T2w MRIs show Mbnl1/2 genotype-dependent regional volume alterations. a, c, e, g** A montage of slices of extracted whole brains of representative 2-mo *Mbnl1$^{+/+}$/Mbnl2$^{+/+}$* (wild-type control) *Mbnl1$^{-/-}$*, *Mbnl2$^{-/-}$*, and *Mbnl1$^{-/-}$/Mbnl2$^{+/-}$* mice that underwent in vivo 2D FSE T2w MRIs, respectively, are shown. **b, d, f, h** Show regions of interest (ROI) manually drawn and color-filled as described in the Methods. Only ROIs that are significantly different from the control *Mbnl1$^{+/+}$/Mbnl2$^{+/+}$* brains are color-filled in the *Mbnl1$^{+/+}$/Mbnl2$^{+/+}$*, *Mbnl1$^{-/-}$*, *Mbnl2$^{-/-}$*, and *Mbnl1$^{-/-}$/Mbnl2$^{+/-}$* montages. Colored regions are white matter regions, which include the anterior commissure and corpus callosum/external capsule (yellow), and gray matter regions, which include the cerebellum (red), the hippocampus (green), and the hindbrain (pink). Calculated areas from the ROIs were multiplied by the image-slice thickness to generate ROI volumes shown in Figs. 5–7. The olfactory bulb was excluded in volume measurements. Slice positions were identified based on Bregma locations. The slice where the anterior commissure is connected between hemispheres approximates Bregma, which is defined as 0.0 mm. Going from rostral to caudal slice positions, boxes indicate structural landmarks that include Bregma (red box), the last slice containing hippocampal regions (green box), and the end of the brain (yellow box). Slice thickness = 0.5 mm. Scale bar = 1 mm.

months of age in the isocortex [F (3, 16) = 1.327, *p* = 0.3004] (Fig. 7a; Table 2). A significant group effect was observed in the hippocampus volumes [F (3, 16) = 5.435, *p* = 0.0090]. A post hoc Dunnett's multiple-comparison test showed that the hippocampus volume was comparably reduced in the *Mbnl1$^{-/-}$* (~13% decrease, *p* = 0.0214), in the *Mbnl2$^{-/-}$* (~14% decrease, *p* = 0.0138) and in the *Mbnl1$^{-/-}$/Mbnl2$^{+/-}$* mice (~15% decrease, *p* = 0.0072) (Fig. 7b, g; Table 2). No difference was observed in the caudoputamen region [F (3, 16) = 0.4026, *p* = 0.7531] (Fig. 7c; Table 2).

In the brainstem, the inter-/midbrain volumes showed no difference between the 2-month-old cohorts [F (3, 16) = 1.120, *p* = 0.3703] (Fig. 7d; Table 2). A group effect was observed by ANOVA analysis in the hindbrain volumes [F (3, 16) = 7.054,

*p* = 0.0031], with the post hoc Dunnett's test showing reduced volumes in *Mbnl1$^{-/-}$* (~12% decrease, *p* = 0.0431) and in *Mbnl1$^{-/-}$/Mbnl2$^{+/-}$* (~20% decrease, *p* = 0.0009) brains when compared with age-matched *Mbnl1$^{+/+}$/Mbnl2$^{+/+}$* brains (Fig. 7e, h; Table 2).

An unpaired two-tailed *t*-test showed a significant volume reduction in the cerebellum of the *Mbnl1$^{-/-}$/Mbnl2$^{+/-}$* mice (~9% decrease, *p* = 0.044) but not in the *Mbnl1$^{-/-}$* or the *Mbnl2$^{-/-}$* mice, when compared with age-matched *Mbnl1$^{+/+}$/Mbnl2$^{+/+}$* control mice (Fig. 5f, i; Table 2). In summary, this analysis demonstrates Mbnl1- and Mbnl2-specific effects in initiating region-specific volume alterations in the adult mouse brain (Fig. 8a, g). Western blot analysis of Mbnl1 and Mbnl2 levels in the cortex, hippocampus, cerebellum, hind and midbrain regions, and all other regions,

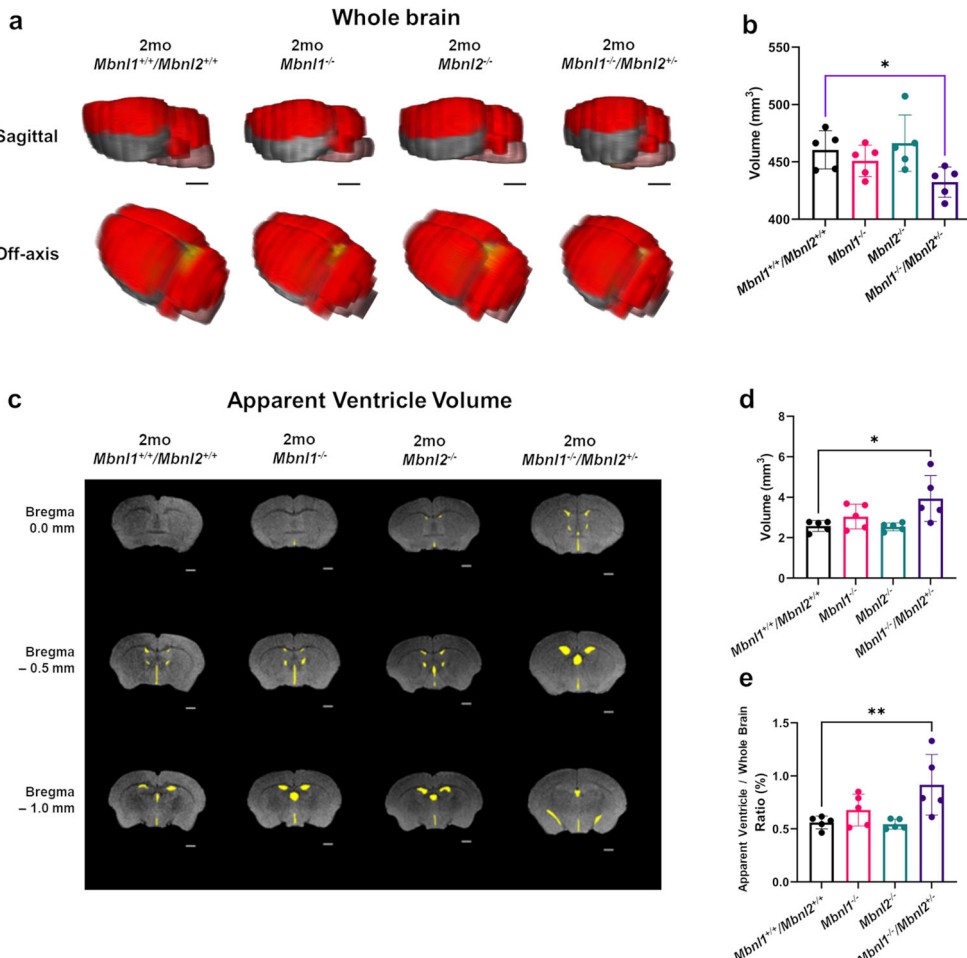

**Fig. 5 Two-month *Mbnl1*$^{-/-}$/*Mbnl2*$^{+/-}$ mice exhibit reduced whole-brain volume and enlarged ventricles. a** 3D rendering of representative whole-brain 2D FSE T2w MRIs for 2-mo *Mbnl1*$^{+/+}$/*Mbnl2*$^{+/+}$, *Mbnl1*$^{-/-}$, *Mbnl2*$^{-/-}$, and *Mbnl1*$^{-/-}$/*Mbnl2*$^{+/-}$ mice are shown. Colored regions are isocortex and cerebellum (red), hindbrain (pink), white matter areas (yellow), and olfactory gray matter areas (gray). The olfactory bulb was excluded in volume measurements. Scale bar = 2.0 mm. **b** Whole-brain measurements (mean ± SD) are shown. **c** Representative axial brain slices from FSE T2w MRI scans where apparent ventricle volumes were defined from hyperintense regions along ventricle locations (yellow filled). Scale bar in 2D axial images denotes 1 mm. **d**, **e** Apparent ventricle volume and percent apparent ventricle volume/whole-brain volume (mean ± SD) are shown. Black brackets: Dunnett's test; Purple brackets: two-tailed *t*-test. Error bars are standard deviations (SD). In all cases, *$p < 0.05$, **$p < 0.01$.

including the interbrain and caudoputamen in *Mbnl1*$^{+/+}$/*Mbnl2*$^{+/+}$ brains, demonstrated Mbnl1 and Mbnl2 expression in all regions studied. Compensatory increases in Mbnl2 and Mbnl1 were observed in all *Mbnl1*$^{-/-}$ and *Mbnl2*$^{-/-}$ brain regions examined, respectively, with the exception of the cortex (Fig. 8b–f).

## Discussion

The results from this study support the hypothesis that CUG-expansion RNA- mediated stochastic losses in MBNL1 and MBNL2 are an important driver of the structural alterations reported in DM1 patient brains. Consistent with CUG- repeat length playing a regulatory role in the severity of the structural brain defects in DM1, congenital DM1 patients, who tend to have the largest CTG expansions[20], show severe brain pathology manifesting as microcephaly, whole-brain volume reductions, and enlarged ventricles[3,4,22,23]. As Mbnl2 levels are greater than Mbnl1 levels in the brain[9], *Mbnl1*$^{+/-}$/*Mbnl2*$^{-/-}$ brains are predicted to have the largest overall reduction in Mbnl1 and Mbnl2 levels when compared with *Mbnl1*$^{-/-}$ and *Mbnl2*$^{-/-}$ and *Mbnl1*$^{-/-}$/*Mbnl2*$^{+/-}$ brains. In this study, 4-month-old *Mbnl1*$^{+/-}$/*Mbnl2*$^{-/-}$ mice show the largest decrease in whole-

brain volumes (~9%) and increased apparent ventricle volume (~126% increase) and apparent ventricle volume to whole-brain volume ratio (~152% increase) when compared with controls. In the 2-month-old cohorts studied, *Mbnl1*$^{-/-}$/*Mbnl2*$^{+/-}$ mice, but not the *Mbnl1*$^{-/-}$ and the *Mbnl2*$^{-/-}$ mice, show alterations in whole-brain (~6% decrease) and ventricle volumes (~53% increase) (Fig. 8a, g; Tables 1 and 2). Although direct comparisons between the two groups cannot be made, these data demonstrate that dual depletion of Mbnl1 and Mbnl2 initiates brain-volume reductions and ventricular volume increases. Thus, it is likely that the increasing capacity of larger CUG-expansion RNAs to sequester and inactivate the MBNL1 and MBNL2 proteins is an important driver of both whole-brain volume decreases and ventricular enlargement in congenital DM1. It is not known if the enlarged ventricles in *Mbnl1*$^{+/-}$/*Mbnl2*$^{-/-}$ and *Mbnl1*$^{-/-}$/*Mbnl2*$^{+/-}$ mice reflect brain-volume shrinkage or if these events are separable pathologies.

A second prediction implicit in this hypothesis is that brain defects manifesting in DM1 patients encoding both small and large CTG tracts must be sensitive to smaller reductions in MBNL1 and MBNL2 levels. White matter defects are a widely reported pathology in both adult-onset DM1 patients, who have

**Table 2 Summary of brain region volumes (in mm³) from 2-month-old Mbnl1+/+/Mbnl2+/+, Mbnl1−/−, Mbnl2−/−, and Mbnl1−/−/Mbnl2+/− mice.**

| Region | 2mo Mbnl1+/+/Mbnl2+/+ | | 2mo Mbnl1−/− | | | | 2mo Mbnl2−/− | | | | 2mo Mbnl1−/−/Mbnl2+/− | | | |
|---|---|---|---|---|---|---|---|---|---|---|---|---|---|---|
| | Mean | SD | Mean | SD | Change (percent) | p-value (Dunnett's) | Mean | SD | Change (percent) | p-value (Dunnett's) | Mean | SD | Change (percent) | p-value (Dunnett's) |
| Anterior commissure | 0.89 | 0.06 | 0.78 | 0.06 | −12.36 | 0.1763 | 0.61 | 0.08 | −31.46 | 0.0008 | 0.63 | 0.11 | −29.21 | 0.0015 |
| Corpus callosum/external capsule | 10.44 | 1.43 | 7.45 | 0.96 | −28.64 | 0.001 | 7.17 | 0.72 | −31.32 | 0.0004 | 5.97 | 0.96 | −42.82 | 0.0001 |
| Caudoputamen | 18.23 | 1.51 | 18.23 | 0.81 | 0 | 0.9999 | 17.68 | 0.7 | −3.02 | 0.7521 | 18.35 | 1.08 | 0.66 | 0.9958 |
| Cerebellum | 80.66 | 6.08 | 75.19 | 4.12 | −6.78 | 0.3275 | 80.36 | 7.89 | −0.37 | 0.9996 | 73.05 | 3.71 | −9.43 | 0.1234 / 0.044 (t-test) |
| Isocortex | 100.71 | 2.92 | 104.24 | 4.03 | 3.51 | 0.3793 | 99.82 | 3.52 | −0.88 | 0.9698 | 102.99 | 5.04 | 2.26 | 0.6922 |
| Isocortex: Bregma+[a] | 36.96 | 2.65 | 34.14 | 4.15 | −7.63 | 0.3371 | 35.88 | 2.56 | −2.92 | 0.8912 | 38.55 | 2.12 | 4.3 | 0.7347 |
| Isocortex: Bregma-[b] | 63.75 | 3.98 | 70.1 | 1.53 | 9.96 | 0.0149 | 63.94 | 2.82 | 0.3 | 0.9993 | 64.44 | 3.65 | 1.08 | 0.9717 |
| Hindbrain | 53.82 | 5.86 | 47.45 | 2.31 | −11.84 | 0.0431 | 48.99 | 2.98 | −8.97 | 0.1435 | 42.94 | 2.92 | −20.22 | 0.0009 |
| Hippocampus | 21.13 | 1.13 | 18.44 | 1.48 | −12.73 | 0.0214 | 18.25 | 1.14 | −13.63 | 0.0138 | 17.97 | 1.78 | −14.96 | 0.0072 |
| Inter/midbrain | 57.78 | 1.64 | 59.1 | 4.24 | 2.28 | 0.9066 | 57.69 | 5.01 | −0.16 | 0.9999 | 54.8 | 3.58 | −5.16 | 0.485 |
| Whole brain | 460.56 | 16.59 | 450.94 | 13.78 | −2.09 | 0.7251 | 466.41 | 24.56 | 1.27 | 0.9136 | 432.39 | 13.21 | −6.12 | 0.0567 / 0.0179 (t-test) |
| Apparent ventricles[c] | 2.58 | 0.27 | 3.05 | 0.61 | 18.22 | 0.5622 | 2.54 | 0.19 | −1.55 | 0.9993 | 3.94 | 1.14 | 52.71 | 0.014 |
| Apparent ventricle/whole brain ratio (%) | 0.56 | 0.06 | 0.68 | 0.15 | 21.43 | 0.5531 | 0.54 | 0.05 | −3.57 | 0.9976 | 0.92 | 0.29 | 64.29 | 0.0099 |
| Number of mice per genotype | 5 | | 5 | | | | 5 | | | | 5 | | | |

relatively small CTG tracts, and in congenital DM1 patients, who tend to have longer CTG tracts[20]. White matter defects manifest as hyperintensities, microstructural damage, and volume reductions in T2-weighted MRI scans, with some regions, such as the anterior temporal lobe and the corpus callosum, being more frequently affected[4,24,27]. White matter defects can be particularly severe in congenital DM1, where prominent thinning of the corpus callosum is reported[4,23]. Longitudinal studies and disease-duration-matched cohorts show an increase in white matter defects with aging, suggesting that somatic CTG expansions can lead to progressively larger sequestrations of MBNL1 and MBNL2 with time[23,25,26]. Commensurate with these observations, we show comparable changes in the corpus callosum/external capsule volumes in 2-month-old Mbnl1−/− mice (~29% decrease) and Mbnl2−/− mice (~31% decrease). Two-month-old Mbnl1−/−/Mbnl2+/− mice show a more prominent defect, with an ~43% decrease in the corpus callosum/external capsule volume (Fig. 8a, g; Table 2). Thus, the loss of either Mbnl1 or Mbnl2 per se, can result in striking white matter volume reductions. Consistent with the prominent corpus callosum thinning in congenital DM1 and the increased severity of white matter defects with aging, 2-month-old Mbnl1−/−/Mbnl2+/− mice show a more severe reduction (~43% decrease) in the corpus callosum/external capsule volume when compared with control mice. These observations are consistent with a previous study demonstrating a role for cytoplasmic Mbnl1 in neurite outgrowth and development[33]. Four-month-old Mbnl1+/−/Mbnl2−/− mice show a less severe change (~21% decrease) in the corpus callosum/external capsule volume. It is unclear if this decrease in severity reflects possible compensation occurring in the corpus callosum/external capsule with age.

In contrast to the corpus callosum, in the anterior commissure, which is a white matter tract connecting the two temporal lobes, gene-specific effects on volume alterations are observed. Specifically, the anterior commissure shows greater sensitivity to Mbnl2 loss, as 2-month-old Mbnl2−/− mice show a significant decrease in anterior commissure volumes (~31%), while 2-month-old Mbnl1−/− mice do not. Two-month-old Mbnl1−/−/Mbnl2+/− mice show a comparable phenotype to Mbnl2−/− mice, with (~29% decrease). Consistent with the stronger requirement for Mbnl2 in maintaining the structural integrity of the anterior commissure, 4-month-old Mbnl1+/−/Mbnl2−/− mice show a severe loss in anterior commissure volumes when compared with controls (~49%) (Fig. 8a, g, Tables 1 and 2). Thus, stochastic losses in the MBNL1 and MBNL2 proteins, concurrent with gene and dose-specific effects in white matter tracts, can give rise to the variability in white matter defects reported in DM1 patients[2–4]. We do not observe white matter hyperintensities in Mbnl1- and Mbnl2- deficient mice. It is not clear if this absence reflects human–mouse differences or if hyperintensities in patients result from other mechanisms.

Gray matter defects are observed frequently in the cortex, hippocampus, basal ganglia, striatum, thalamus, nucleus accumbens, cerebellum, and diencephalon in DM1 patients[4]. Compound losses in Mbnl1 and Mbnl2 result in volume shrinkages in all gray matter regions that were measured in this study, with the exception of the caudoputamen in Mbnl1+/−/Mbnl2−/− mice. Specifically, 4-month-old Mbnl1+/−/Mbnl2−/− mice show volume reductions in the isocortex (~8%), the hippocampus (~19%), the inter-/midbrain (~15%), and the hindbrain (~18%).

Gene-specific effects are observed in the 2-month-old cohorts, with volume decreases in the cerebellum (~9%) requiring both the loss of Mbnl1 and the partial depletion of Mbnl2. Hindbrain integrity is more sensitive to reductions in Mbnl1, as only Mbnl1−/− (~12%) and not Mbnl2−/− mice show significant hindbrain-volume decreases. Strikingly, hippocampus volumes

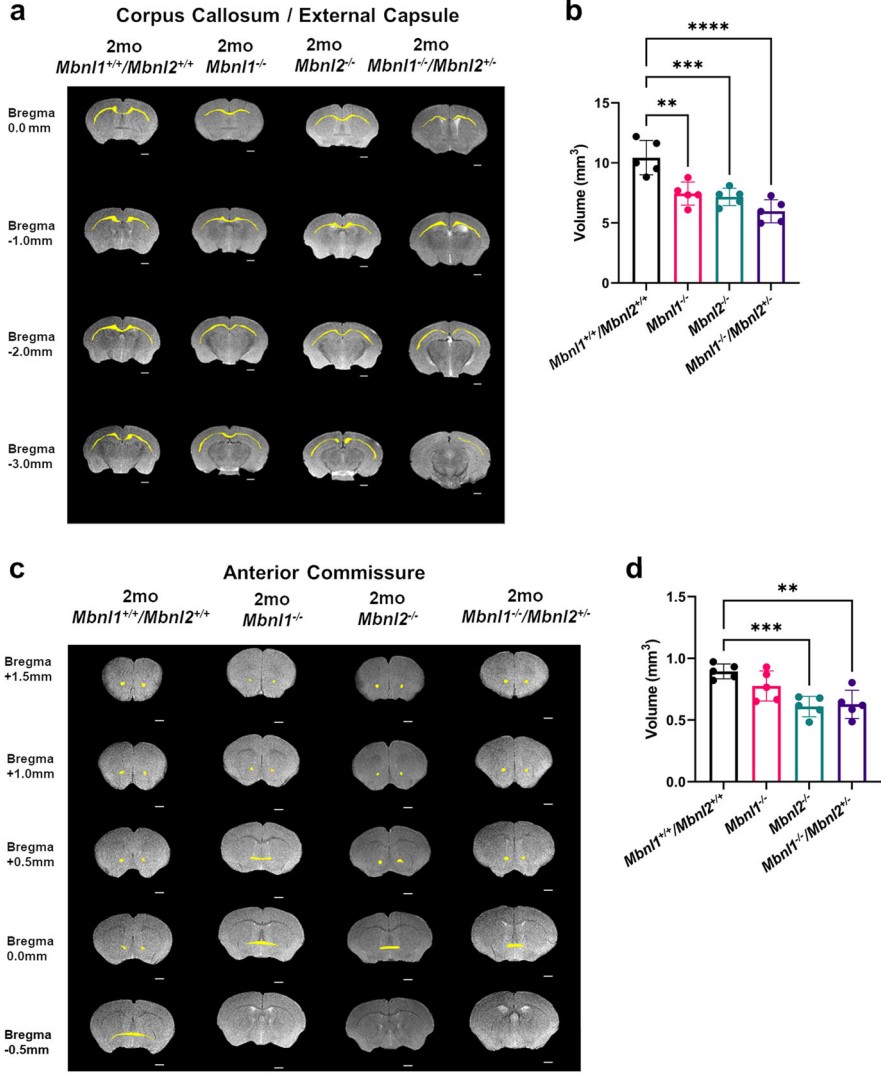

**Fig. 6 White matter regional volume decreases in 2mo *Mbnl1*⁻/⁻, *Mbnl2*⁻/⁻, and *Mbnl1*⁻/⁻/*Mbnl2*⁺/⁻ mice. a** Representative axial brain slices from FSE T2w MRI scans with the corpus callosum/external capsule volumes overlaid in yellow. **b** Corpus callosum/external capsule volume measurements (mean ± SD) are shown. **c** Representative axial brain slices from FSE T2w MRI scans with the anterior commissure volumes overlaid in yellow. **d** Anterior commissure volume measurements (mean ± SD) are shown. Scale bar in 2D axial images denotes 1 mm. Error bars are standard deviations (SD). In all cases, $*p < 0.05$, $**p < 0.01$, $***p < 0.001$, and $****p < 0.0001$.

appear to be very susceptible to the loss of either Mbnl1 or Mbnl2 and all three 2-month-old cohorts, *Mbnl1*⁻/⁻ (~13%), *Mbnl2*⁻/⁻ (~14%), and *Mbnl1*⁻/⁻/*Mbnl2*⁺/⁻ (~15%) show comparable hippocampal-volume losses. Thus, Mbnl1/2 depletion results in widespread gray matter volume reductions in the mouse brain in a manner reminiscent of DM1 patients (Fig. 8a, g, Tables 1 and 2).

Isocortex-volume decreases (~8%) are observed only in *Mbnl1*⁺/⁻/*Mbnl2*⁻/⁻ mice. This brain region is therefore more resistant to Mbnl1- and Mbnl2-dosage changes and appears to require a substantial decrease in total Mbnl1 and Mbnl2 levels to result in shrinkage. As cortex-volume decreases are observed in DM1 patients with small and large CTG tracts[3,4,24], this observation suggests that alternate mechanisms may work with Mbnl1 and Mbnl2 loss to enhance cortex-volume reductions in DM1. Similarly, although gray matter atrophy is reported in the caudate nucleus and putamen in DM1 patients[24], we observe an increase of ~7% in the caudoputamen in *Mbnl1*⁺/⁻/*Mbnl2*⁻/⁻ mice. Other CUG-expansion RNA-mediated mechanisms must therefore be important in the development of pathology in this region.

It is debated if structural alterations in the brain contribute to behavioral changes in DM1 patients[2,4]. Impaired visuospatial learning and memory is pervasive in DM1 patients encoding a wide range of repeat sizes[2,24,34–37]. Consistent with this observation, we observe that the hippocampus, a region that contains a cognitive map of space, and which plays a role in learning and memory[38,39], is particularly sensitive to alterations in Mbnl1 and Mbnl2 dosage. Thus, correlative changes in hippocampal-volume decreases and impaired visuospatial learning and memory are observed in the *Mbnl1*⁻/⁻ and the *Mbnl2*⁻/⁻ mice[29,30]. White matter volume decreases are observed in *Mbnl1*⁻/⁻ brains and increase in prominence in *Mbnl2*⁻/⁻, *Mbnl1*⁻/⁻/*Mbnl2*⁺/⁻, and *Mbnl1*⁺/⁻/*Mbnl2*⁻/⁻ brains (Fig. 8a, g, Tables 1 and 2). Loss of structural and functional connectivity between gray matter regions arising from white matter loss may impact behavior in DM1 in ways that have yet to be fully understood.

Our results demonstrate that Mbnl1 and Mbnl2 gene- and dose-specific effects impact mouse brain integrity (Fig. 8a, g). Such effects, in conjunction with the stochastic nature of the sequestration of the MBNL1 and MBNL2 proteins by CUG-

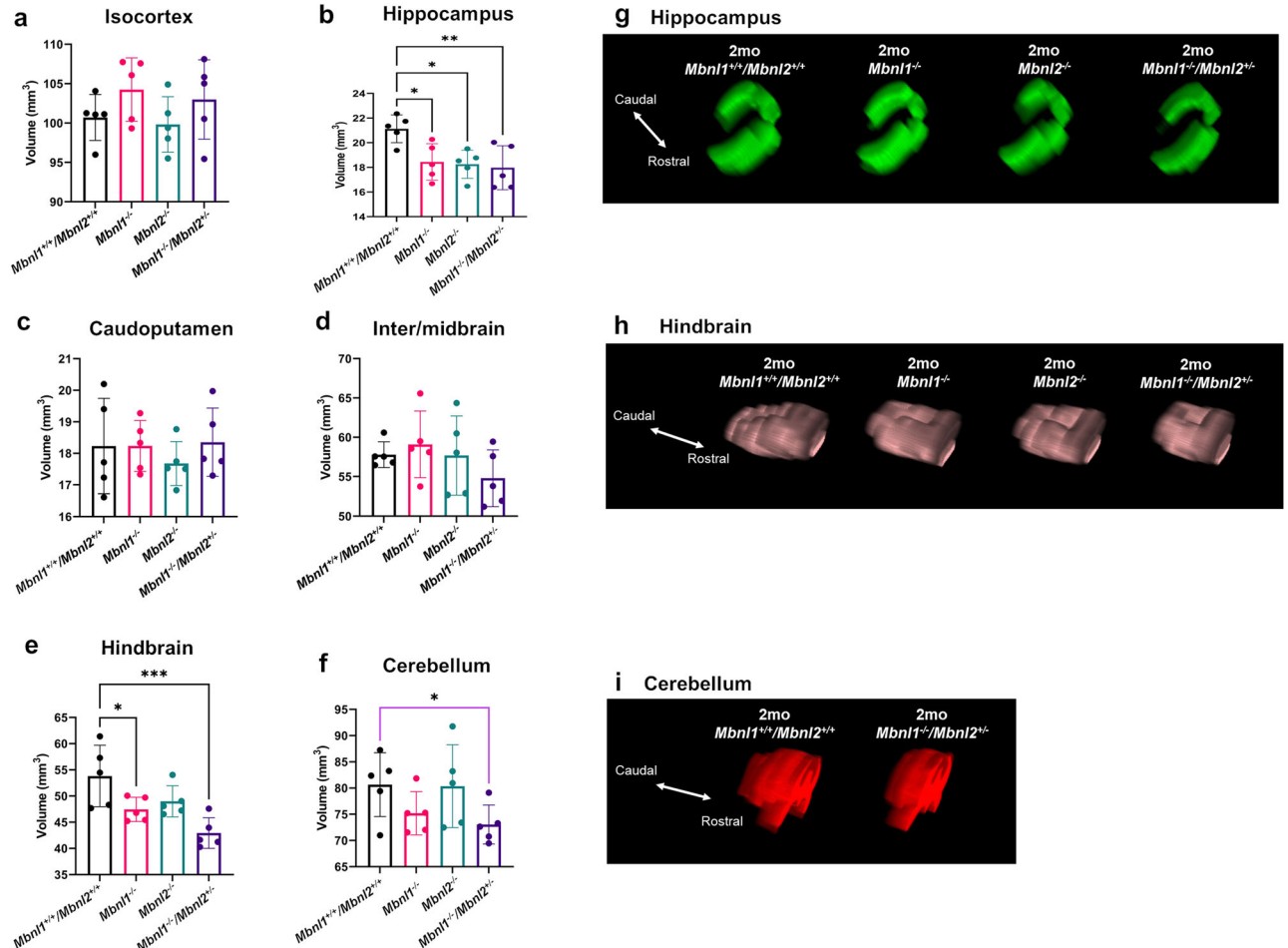

**Fig. 7 Gray matter regional volume decreases in 2mo *Mbnl1*⁻/⁻, *Mbnl2*⁻/⁻, and *Mbnl1*⁻/⁻/*Mbnl2*⁺/⁻ mice. a–f** Isocortex, hippocampus, caudoputamen, inter-/midbrain, hindbrain, and cerebellum volume measurements (mean ± SD) are shown. **g–i** 3D caudal–rostral projections of representative hippocampi, hindbrains, and cerebella are shown. Black brackets: Dunnett's test; Purple brackets: two-tailed *t*-test. Error bars are standard deviations (SD). In all cases, *$p < 0.05$, **$p < 0.01$, and ***$p < 0.001$.

expansion RNAs and the intergenerational and somatic expansions of CTG repeats in DM1 brains, can explain the highly variable structural brain alterations reported in DM1 patients and phenotype intensification in congenital DM1 brains[1–4]. Region-specific gray matter volume shrinkages and loss of connectivity resulting from white matter defects may cooperate with functional alterations in Mbnl1/2- target RNAs to impact behavior in DM1 patients. As the structural brain alterations in congenital DM1 that manifest at birth are likely to be developmental in origin and because DM1 structural brain defects can progress with time in all three classes of patients[22,23,25,26], our data support that displacement of MBNL1 and MBNL2 from CUG repeat-expansion RNAs begins *in utero* and continues postnatally to prevent brain-volume shrinkages and potentially behavioral alterations in DM1.

## Methods

**Animals**. Mice were housed in standard cages on a 12 h light/dark cycle with access to water and a standard laboratory diet *ad libitum*. We have previously described the development of 129sv *Mbnl1*^ΔE2/ΔE2^ mice (indicated as *Mbnl1*⁻/⁻ in this study) where the ATG encoding Mbnl1 exon 2 is deleted[32]. Development of 129sv *Mbnl2*^ΔE2/ΔE2^ mice (indicated as *Mbnl2*⁻/⁻) where the ATG encoding Mbnl2 exon 2 is deleted, is described in Supplementary Methods and shown in Supplementary Fig. 1. These strains were bred to develop 129sv *Mbnl1*^ΔE2/ΔE2^/*Mbnl2*⁺/ΔE2 and 129sv *Mbnl1*⁺/ΔE2/*Mbnl2* ^ΔE2/ΔE2^ mice denoted here as *Mbnl1*⁻/⁻/*Mbnl2*⁺/⁻ and *Mbnl1*⁺/⁻/*Mbnl2*⁻/⁻ mice, respectively. Age- and gender-matched wild-type 129sv mice (indicated as *Mbnl1*⁺/⁺/*Mbnl2*⁺/⁺) were used as controls. *Mbnl1*⁺/⁺/*Mbnl2*⁺/⁺, *Mbnl1*⁻/⁻, *Mbnl2*⁻/⁻, *Mbnl1*⁺/⁻/*Mbnl2*⁻/⁻, and *Mbnl1*⁻/⁻/*Mbnl2*⁺/⁻ 129sv female mice at 2 and 4 months of age were used in this study. We have complied with the relevant

ethical regulations for animal testing and research as specified by protocol number 11970, approved by the Institutional Animal Care and Use Committee of the University of Southern California.

**In vivo MRI**. All MRI scans were performed using an MR Solutions (Guildford, UK) 7T MRI-PET scanner using a 20 mm internal-diameter quadrature bird cage mouse head coil. The scanner is capable up to 7T and has ~24 cm bore size, with up to 600 mT/m maximum gradient. Mice were anesthetized with 2% isoflurane at flow rate 250 μL/min using (Somnosuite, Kent Scientific). Animals were transferred onto a temperature controlled heated scanner bed (Minerve, France) with anesthesia levels maintained between 1.5–2%. Temperature was monitored using a fiber-optic thermocouple and maintained at 37 °C. A pneumatic pillow was placed underneath the animal for respiration monitoring. Mice were positioned at the magnet-bore isocenter using a motorized system. Temperature and respiration were monitored by the accompanying PC-SAM software (SAII, USA). Animals were placed in a heated recovery chamber after the scan, until the animals were ambulatory. A gradient echo scan was used to obtain three orthogonal slices for positioning followed by 2-dimensional (2D) fast spin-echo (FSE) T2-weighted (T2w) sequences to define neuroanatomy in the transverse axial orientation. FSE T2w scan parameters were as follows: TE = 45 ms, TR = 4000 ms, number of averages = 4, echo-train length = 7, field of view = 14 mm × 14 mm, slice thickness = 0.5 mm, number of slices = 28, and matrix size = 256 × 256.

**Image analysis**. Regions of interest (ROI) were manually defined on the 2D T2w FSE MRI scans using the polygon tool in ImageJ software[40] and using Hof et al.[41] and the Allen mouse brain atlas as visual guides (http://mouse.brain-map.org/). ROI area measurements for each slice were multiplied by the image-slice thickness (0.5 mm) and summed to generate volumes. Three blinded experimenters (N.S.S., C.Z., and S.J.L.) independently drew all ROIs and calculated volumes. Statistical analysis of volume measurements demonstrated no significant differences between the experimenters. Discussions were held by the experimenters to reach an unequivocal consensus on

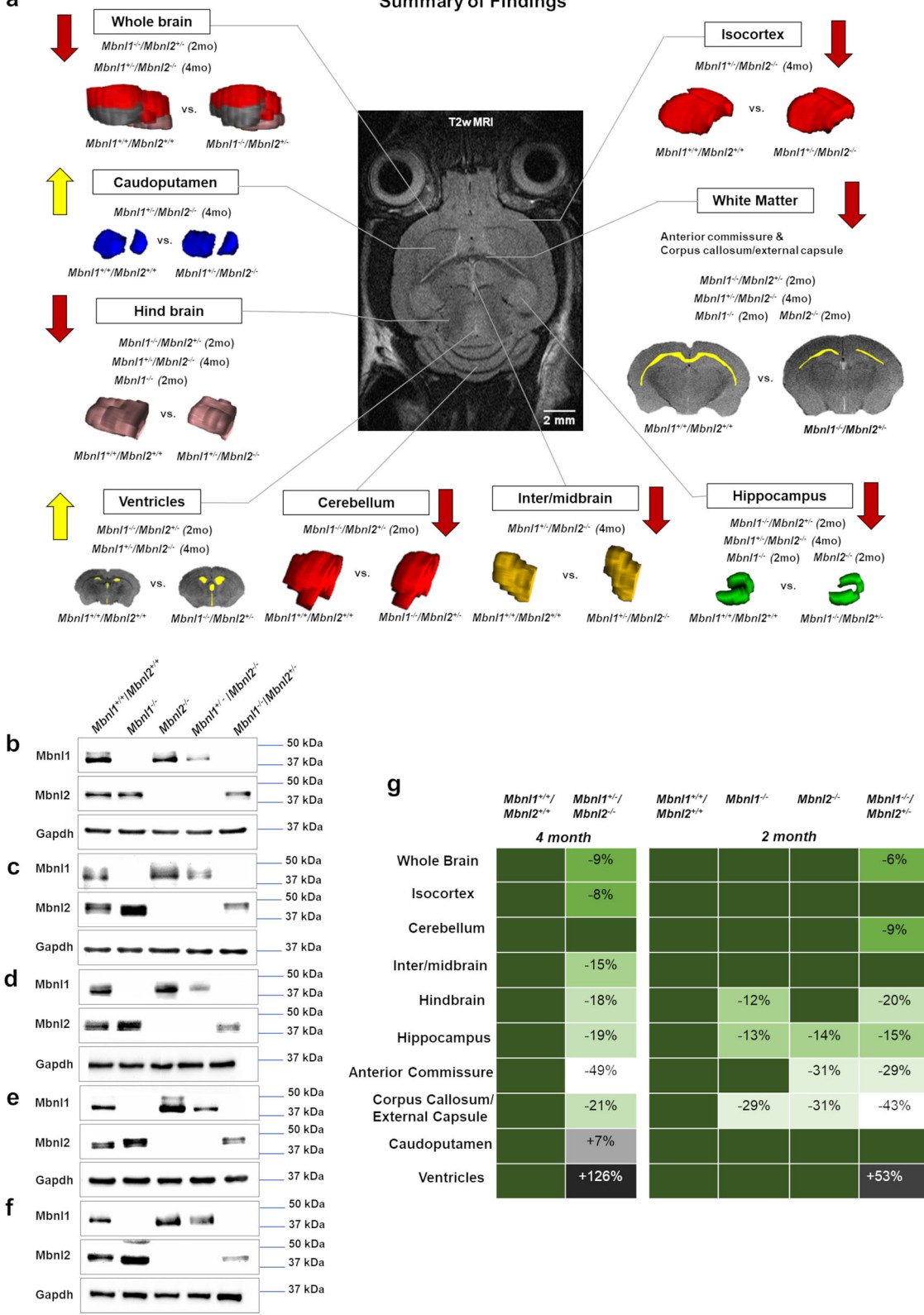

**Fig. 8 Summary. a** Region-specific alterations in Mbnl1 and Mbnl2 deficient mice are shown. **b–f** Mbnl1- and Mbnl2- levels measured by western blot analyses in the cortex (**b**), hippocampus (**c**), cerebellum (**d**), mid- and hindbrain (**e**), and all other brain regions, including the interbrain and the caudoputamen (**f**), are shown. Gapdh was used as an internal control. **g** Mbnl1/2 genotype-specific effects on brain regional volume alterations are shown. Increasingly severe volume decreases are indicated in paler- green shades and volume increases are indicated in darker-gray and black tones.

disparate measurements. The following white and gray matter ROIs were measured: defined white matter regions included the anterior commissure and the corpus callosum/external capsule. Gray matter regions included the cerebrum (isocortex, hippocampus proper and subiculum, denoted as hippocampus and caudoputamen), brainstem (inter-/midbrain and hindbrain), and the cerebellum. Whole-brain and apparent ventricle volumes were measured. Apparent ventricle volumes were delineated as visibly identifiable ventricle regions with hyperintense signals in the FSE T2w brain slices. The olfactory bulb was excluded in volume measurements.

**Western blot analysis**. Cortex, hippocampus, cerebellum, mid-/hindbrain regions, and all other brain regions, including interbrain and caudoputamen, were dissected and flash-frozen in liquid nitrogen. Frozen brain samples were lysed with modified RIPA buffer (50 mM Tris-HCl, pH 7.4; 1% NP40; 0.25% Na-deoxycholate; 150 mM NaCl; 1 mM EDTA with 1X protease-inhibitor cocktail; 1 mM NaF; 1 mM NaVO4; 1 mM PMSF) and equal amounts of protein (15 ug) were separated by 10% SDS-PAGE and transferred to nitrocellulose membranes. Membranes were blocked with 5% skim milk in TBST (1 M Tris-HCl, pH 7.5; 5 M NaCl; 10% Tween20), for 1 h at room temperature and incubated with primary antibodies at 4 ℃ overnight with rocking. The membranes were subsequently incubated for 2 h at room temperature with secondary antibodies conjugated with horseradish peroxidase. After washing with TBST, the membranes were incubated with Pierce$^{TM}$ ECL plus western blotting substrate (Thermo Scientific) according to the manufacturer's protocol and the signal was detected on a Chemi$^{TM}$ Doc MP imaging system (Bio-Rad). Anti-Gapdh and anti-Mbnl2 antibodies were purchased from Santa Cruz Biotechnology (Gapdh, Cat # sc-32233, Lot #K3016, 100 ug/ml, dilution 1:200; Mbnl2 Cat # sc-136167, Lot # D0417, 200 ug/ml, dilution 1:200 (monoclonal antibody raised against recombinant human Mbnl2)), anti-Mbnl1 antibodies are a gift from Ian Holt, dilution 1:100[42]. Uncropped Western blot images are shown in Supplementary Fig. 2a–e.

**Statistics and reproducibility**. Two tailed t-test was performed when comparing the ROI volumes (mean ± SD) of 4-month-old $Mbnl1^{+/+}/Mbnl2^{+/+}$ and $Mbnl1^{+/-}/Mbnl2^{-/-}$ cohorts. One-way ANOVA analysis was performed when comparing the ROI volumes (mean ± SD) of 2-month-old $Mbnl1^{+/+}/Mbnl2^{+/+}$, $Mbnl1^{-/-}$, $Mbnl2^{-/-}$, and $Mbnl1^{-/-}/Mbnl2^{+/-}$ mice, followed by the Dunnett's multiple-comparison test. In some cases, the two-tailed t-test was performed when comparing genotypes in the 2-month-old cohorts. GraphPad Prism 9 (San Diego, CA, USA) statistical software was used and a value of $p < 0.05$ was defined as statistically significant. For all 4- and 2-month cohorts, $n = 5$ mice/genotype.

**Reporting summary**. Further information on research design is available in the Nature Research Reporting Summary linked to this article.

## Data availability

All source data are shown in Supplementary Figs. 2–8 and Supplementary Data 1. No custom codes or mathematical algorithms were used in these analyses. Transgenic animals are available for noncommercial purposes from S.R.

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

## Acknowledgements

The authors declare that they have no conflict of interest. Imaging was carried out in the Functional Biological Imaging Core at the USC Zilkha Neurogenetic Institute. This study was supported in part by R01NS080547 to SR and R03NS107724 to LC.

## Author contributions

N.S.S., C.Z., S.J.L., and S.R. collected and analyzed the MRI data, P.V. carried out the Western blot analysis, X. Li and J.C. developed the mouse cohorts and collected the MRI data, X. Liu and R.J. collected MRI data, and S.R. and L.C supervised the study. N.S.S., C.Z., and S.R. wrote the paper. S.R. designed the study.

## Competing interests

The authors declare no competing interests.
