## [Transparent Peer Review File · Communications Biology]

Reviewers' comments:

Reviewer #1 (Remarks to the Author):

Using MBNL1 and 2 knockout mouse models, Maria et al present the findings of MBNL protein family regulates brain structure integrity. In the present study, loss of MBNL in brains of *Mbnl1+/-/Mbnl2-/-* female mice resulted in ~10% of reduction in total brain volume and enlargement of ventricle. Reduction in certain brain region may account for the reduced brain volume including isocortex, hippocampus, inter/midbrain and hindbrain. To further determine how individual MBNL members affected the brain structure, the authors found loss of either MBNL1 or MBNL2 alone did not significantly affect brain size, however *Mbnl1-/-/Mbnl2+/-* brains exhibited similar extent of brain volume reduction (~10%) to that seen in *Mbnl1+/-/Mbnl2-/-* brain. The authors conclude that loss of MBNL protein family impairs brain structure integrity may occurs early during embryonic stage and continuously affect brain structure and behaviors throughout adulthood.

The authors of the present study used MBNL1/2 knockout mice to investigate how and whether MBNL loss of function to impair brain structure and its correlation to DM1 brain features. Loss of MBNL1 has been implicated in affecting neurite outgrowth, therefore may also affecting brain structure. However, it is still unknown the causal mechanism. Although whether loss of MBNL2 also affects brain structure has not been characterized, it has been shown that MBNL2 knockout mice exhibited postnatal growth delay including brain. Therefore, the current study is not conceptually advanced for providing novel knowledge to understand the pathogenesis of DM1 brain.

There are several major concerns about the characterization of the mouse models as well as the interpretation of the results.

1. The MBNL2 KO model used in the current study was newly established. However, (a) there is no description about what Cre line was used for generating MBNL2 KO model; (b) there were no data showing loss of MBNL2 in the brain. It looks like that authors assumed their KO model was similar to what Swanson lab has established. If so, MBNL1 level was increased in Swanson's MBNL2 KO brain. How about the MBNL1 level in this newly generated MBNL2 KO brain?

2. It is not known or shown whether MBNL1 and 2 are co-expressed in the same brain regions or exclusively distributed in different brain regions. Subcellular expression pattern, temporal and spatial expression patterns of MBNL1 and 2 may be different, two KO models are different, one conventional one likely region-specific

Therefore, the basis of comparing loss of MBNL1 and 2 in contributing structural defects becomes very weak. In addition, the knockout efficiency will depend on different Cre line.

In addition, the temporal expression pattern of MBNL1 and 2 might be different, therefore it is hard to justify whether we can say "genotype sensitive" phenotypes.

3. Based on the 9-10% reduction of total brain volume, it is hard to link this result to "congenital" DM1. The authors used 2 and 4 months old of animals for study, whether they observed an association of brain volume reduction with degeneration, since *Mbnl1-/-* and *Mbnl2-/-* mice did not demonstrate a change in whole brain volume or apparent ventricle volume.

Reviewer #2 (Remarks to the Author):

The authors used MRI to describe structural alteration in the brain of mice that did not express functional Muscleblind-like proteins 1 and/or 2 (Mbnl1, 2). Whereas Mbnl1 depleted mice (Mbnl1 Δ E2/ Δ E2) already exist the authors also generated mice lacking Mbnl2 (Mbnl2 Δ E2/ Δ E2). As complete absence of Mbnl1 and 2 (i.e., Mbnl1 Δ E2/ Δ E2/Mbnl2 Δ E2/ Δ E2) is embryonic lethal, the authors used either Mbnl1+/ Δ E2/Mbnl2 Δ E2/ Δ E2 or Mbnl1 Δ E2/ Δ E2/Mbnl1+/ Δ E2 to search for putative additive effects of the two proteins on the brain phenotype.

The main findings of the ms can be summarized as follow: (1) There exist a global brain atrophy in 4-month-old Mbnl1+/ Δ E2/Mbnl2 Δ E2/ Δ E2 mice (when compared to aged matched control mice). This atrophy includes nearly all analyzed structures, excluding the cerebellum, frontal part of the isocortex and the caudoputamen (the latter is even enlarged) and is accompanied by the presence of an enlarged apparent ventricle volume. (2) There also exist a global brain atrophy in 2-month-old Mbnl1 Δ E2/ Δ E2/Mbnl2+/ Δ E2 mice but not in Mbnl1 Δ E2/ Δ E2 or in Mbnl2 Δ E2/ Δ E2 mice. Thus, there is an additive effect of the two proteins on the brain morphology. The overall reduction of the brain volume in comparison to aged matched control mice (which is similar to the reduction in 4-month-old Mbnl1 Δ E2/ Δ E2/Mbnl1+/ Δ E2 mice to their control group, i.e., roughly 10%) is mainly driven by changes in the cerebellum, hindbrain and corpus callosum/external capsule, thus relates to different brain structures. Again, the overall reduction of the brain volume is accompanied by the presence of an enlarged apparent ventricle volume.

Although the study clearly indicates that constitutive absence of Mbnl1 and/or Mbnl2 controls brain morphology, which is of general interest for the understanding of the etiology of Myotonic Dystrophy Type I, do have some concerns about the design of the experiment.

(1) The advantage of MRI is the possibility to perform longitudinal experiments. I wonder why Mbnl1 Δ E2/ Δ E2 and Mbnl2 Δ E2/ Δ E2 mice were only measured at an age of 2 month but not at an age of 4 month. Then, a comparison to Mbnl1+/ Δ E2/Mbnl2 Δ E2/ Δ E2 would also be possible.

(2) Similarly, why did the authors only perform MRI measurements on 4-month-old Mbnl1+/ Δ E2/Mbnl2 Δ E2/ Δ E2 mice (but not when they were only 2-month-old). Of interest would be e.g., the size of the cerebellum at earlier time points.

(3) there were 3 blinded experimenters to analyze the ROI volumes (including discussions to find consensus, which is ok), but only 3 animals per group were measured (in the 2-month-old cohort). Thus, the statistical power of the results is very low. Also for the 4-month-old cohort the measured number of animals (4 control animals, i.e., the animal group that is probably the most abundant one, and 5 animals for the experimental group) is still low.

Based on this, any comparisons (conclusions) between 2-month-old and 4-month-old animals (basically between Mbnl1+/ Δ E2/Mbnl2 Δ E2/ Δ E2 and Mbnl1 Δ E2/ Δ E2/Mbnl2+/ Δ E2) are problematic. At the end, most of the results remain descriptive, without any clear conclusions about the consequence or the underlying cellular mechanisms that eventually lead to the observed abnormalities (except of course that everything is initiated by an altered postnatal RNA splicing pattern caused by the absence of Mbnl1 and 2). One main effect was the observed change in the volume of the corpus callosum/external capsule; thus, I would have expected at least a corresponding DTI measurement (visualizing changes in the structural connectivity) and/or a histological staining to address e.g., the myelination of the axons in order to substantiate this finding. Ideally these measurements should be performed at different time points early (as early as possible after birth) and late (up to 4 month) during the development. This would at least give some hints about the role of Mbnl1/2 for postnatal development (this in turn could then also serve as start point for a subsequent study using conditioned deletion of the proteins at later time points (e.g., after 2 or 4 month)).

Minor points

- The used voxel size is 54.7 x 54.7 x 500 μ m. These "spaghetti"-like voxels are not optimal to prevent volume effects, slice thickness of 300 μ m or less would be more appropriate. An increase in averaging (from 4 - to 8 or 10) and by it an increase in total measuring time should be still tolerable

for the mice.

- Figures 3-5 and 7, y-axes often start with high values (and not at 0), thus these graphs exaggerate the effects. Furthermore, the same values/results are also presented in Table 1 and 2 (thus, Figure 5-7 are redundant to Table 1, 2 and should be moved to the supplemental part).
- Please indicate the body weights of all individual groups, are there differences between individual groups?
- The abstract states: "Correlative changes in brain region integrity with behavioral alterations reported in Mbnl1^{-/-} and Mbnl2^{-/-} mice support the hypothesis that structural defects combine with altered Mbnl1/2 target RNA biology to impact behavior". This is somehow misleading because in the current ms Mbnl1^{-/-} as well as Mbnl2^{-/-} mice only show minor effects on hippocampal volume (which were only evident when an additional statistical analysis was performed) and the ms presents no behavioural data. This statement should only appear in the Results and Discussion part.
- What is the difference between the here used Mbnl2 $\Delta E2/\Delta E2$ mice and the Mbnl2 $\Delta E2/\Delta E2$ knockout mouse used by Charizanis et al doi.org/10.1016/j.neuron.2012.05.029?

Frank Angenstein, RG Functional Neuroimaging, DZNE Magdeburg, Germany

We thank the Reviewers for their evaluation of our manuscript. Responses to all questions raised by Reviewer-1 and Reviewer-2 are detailed below:

Reviewer#1

1. The MBNL2 KO model used in the current study was newly established. However, (a) there is no description about what Cre line was used for generating MBNL2 KO model; (b) there were no data showing loss of MBNL2 in the brain. It looks like that authors assumed their KO model was similar to what Swanson lab has established. If so, MBNL1 level was increased in Swanson's MBNL2 KO brain. How about the MBNL1 level in this newly generated MBNL2 KO brain?

Response-1: Please note that points a+b were addressed in Supplementary Figure-1 of the original submission. The Protamine-Cre transgenic line was used to generate the *Mbnl2*^{-/-} mice and loss of Mbnl2 protein in *Mbnl2*^{-/-} forebrain was shown by western blot analysis (Supplementary Figure 1, Panel e).

The use of Protamine-CRE (expressed in the male germ line) results in a constitutive knockout (42). Specifically, after fertilization, the diploid cell has one mutant and one wildtype *Mbnl2* allele. Thus, all cells of the resulting animal are heterozygous for the *Mbnl2* mutation. In the revised manuscript we show Mbnl2 and Mbnl1 levels in several brain regions, including the cortex, hippocampus, cerebellum, mid and hindbrain regions and all other brain regions including the interbrain and the caudoputamen in wildtype (indicated as *Mbnl1*^{+/+}/*Mbnl2*^{+/+}), *Mbnl1*^{ΔE2/ΔE2} (indicated as *Mbnl1*^{-/-}) and *Mbnl2*^{ΔE2/ΔE2} (indicated as *Mbnl2*^{-/-}), *Mbnl1*^{+/-}/*Mbnl2*^{-/-} and *Mbnl1*^{-/-}/*Mbnl2*^{+/-} mice. As with a previously described *Mbnl2* exon2 deletion (30) a compensatory increase in Mbnl1 is observed with Mbnl2 loss and vice versa in all brain regions studied with the exception of the cortex. Images are shown in Response-2.

2. It is not known or shown whether MBNL1 and 2 are co-expressed in the same brain regions or exclusively distributed in different brain regions.

Response-2: We show that both Mbnl1 and Mbnl2 are expressed in the cortex, hippocampus, cerebellum, mid and hindbrain regions, and in all other brain regions that include the interbrain and caudoputamen. Please see attached images and legend.

Figure 1: Panels A-E: Mbnl1 and Mbnl2 expression in the Cortex, Hippocampus, Cerebellum, Mid and Hindbrain and all other brain regions including the Interbrain and Caudoputamen in 129sv *Mbnl1^{+/+}/Mbnl2^{+/+}*, *Mbnl1^{-/-}*, *Mbnl2^{-/-}*, *Mbnl1^{+/-}/Mbnl2^{-/-}* and *Mbnl1^{-/-}/Mbnl2^{+/-}* mice. Gapdh was used as an internal control. Cropped and uncropped gel images are shown in each Panel.

3. *Subcellular expression pattern, temporal and spatial expression patterns of MBNL1 and 2 may be different, two KO models are different, one conventional one likely region-specific. Therefore, the basis of comparing loss of MBNL1 and 2 in contributing structural defects becomes very weak. In addition, the knockout efficiency will depend on different Cre line. In addition, the temporal expression pattern of MBNL1 and 2 might be different, therefore it is hard to justify whether we can say “genotype sensitive” phenotypes.*

Response-3: Comparison between *Mbnl1^{ΔE2/ΔE2}* and *Mbnl2^{ΔE2/ΔE2}* mice is valid as both strains are constitutive knockouts. Specifically, floxed mouse mutations (*Mbnl1^{fl/fl}* and *Mbnl2^{fl/fl}*) were created on a 129sv genetic background as follows: 129sv embryonic stem cells were genetically modified and implanted into B6 fertilized eggs. The ensuing chimeras were bred to 129sv mice, such that

mutant 129sv gametes in the chimera and the 129sv gametes of the breeder animals allow the development of genetically modified animals on a 129sv background. Both *Mbnl1^{fl/fl}* and *Mbnl2^{fl/fl}* mice were bred to 129sv Protamine-Cre transgenics to constitutively delete exon2 of *Mbnl1* and exon2 of *Mbnl2* respectively (exon-2 of both genes contains the ATG start site). Thus, both strains are constitutive knockouts that were created identically. Therefore, as the genotypes dictate the alternate outcomes observed in this study, this feature is indicated in the manuscript.

6. Based on the 9-10% reduction of total brain volume, it is hard to link this result to “congenital” DM1. The authors used 2 and 4 months old of animals for study, whether they observed an association of brain volume reduction with degeneration, since Mbnl1^{-/-} and Mbnl2^{-/-} mice did not demonstrate a change in whole brain volume or apparent ventricle volume.

Response-6: The prevalence and severity of brain structural alterations is higher in congenital DM1 (cDM1) patients when compared to adult-onset DM1. Specifically, total brain volume reductions, increases in ventricular volumes, gray and white matter defects (with thinning of the corpus callosum being particularly prominent) are more prevalent and/or more severe in cDM1 (3,4,22,23). cDM1 patients in general show the longest CUG repeat expansions (20), which are predicted to allow a more complete sequestration of MBNL1 and MBNL2, when compared to the shorter CUG expansions observed in adult-onset DM1.

Thus, we argue that if the more complete sequestration of MBNL1 and MBNL2 predicted in cDM1 brains initiates a cDM1-like pathology, the following should be observed in mouse models lacking the *Mbnl* proteins: First, single mutations of *Mbnl1* and *Mbnl2* should exhibit fewer cDM1-like features. Second, dual depletion of *Mbnl1*/*Mbnl2* in *Mbnl1^{-/-}Mbnl^{+/-}* and *Mbnl1^{+/-}Mbnl^{-/-}* mice should lead to a greater prevalence of cDM1-like features when compared to the single mutations of *Mbnl1* and *Mbnl2*. Although the 2- and 4-month cohorts cannot be directly compared, the phenotypes analyzed in this study are largely consistent with these predictions. Specifically, *Mbnl1^{-/-}* and *Mbnl2^{-/-}* mice do not show significant alterations in total brain or ventricle volumes. In contrast, dual depletion of *Mbnl1* and *Mbnl2* in *Mbnl1^{+/-}/Mbnl2^{-/-}* and *Mbnl1^{-/-}Mbnl^{+/-}* mice results in brain volume reductions, ventricle volume and ventricle/brain ratio increases (prominent features in cDM1). Lastly, white and grey matter defects are more widespread in the double knockouts when compared to single knockouts. For these reasons the observed brain phenotypes support the hypothesis that the increasing severity of MBNL1 and MBNL2 losses predicted in cDM1 brains can contribute to reductions in brain volumes, increased ventricular volumes and enhanced gray and white matter volume losses observed in these patients.

Reviewer#2

1. The advantage of MRI is the possibility to perform longitudinal experiments. I wonder why *Mbnl1* $\Delta E2/\Delta E2$ and *Mbnl2* $\Delta E2/\Delta E2$ mice were only measured at an age of 2 month but not at an age of 4 month. Then, a comparison to *Mbnl1* $^{+}/\Delta E2$ /*Mbnl2* $\Delta E2/\Delta E2$ would also be possible.

Response-1: *Mbnl1* $^{-}/Mbnl1$ $^{-}$ mice do not survive beyond 8-9 weeks of age and there is significant attrition of *Mbnl1* $^{-}$ mice between 2-4 months of age due to sudden cardiac death. Therefore, a comparative study was carried out at 2 months of age in wildtype (indicated as *Mbnl1* $^{+}/Mbnl2$ $^{+}$), *Mbnl1* $\Delta E2/\Delta E2$ (indicated as *Mbnl1* $^{-}$) and *Mbnl2* $\Delta E2/\Delta E2$ (indicated as *Mbnl2* $^{-}$) and *Mbnl1* $^{-}/Mbnl2$ $^{+/-}$ mice.

2. Similarly, why did the authors only perform MRI measurements on 4-month-old *Mbnl1* $^{+}/\Delta E2$ /*Mbnl2* $\Delta E2/\Delta E2$ mice (but not when they were only 2-month-old). Of interest would be e.g., the size of the cerebellum at earlier time points.

Response-2: 129sv *Mbnl1* $^{+}/Mbnl2$ $^{-}$ genotypes are rare in our colony. Gender matched *Mbnl1* $^{+}/Mbnl2$ $^{-}$ mice occur at frequency of ~1 per 80 pups, even with the most optimum breeding strategies as determined experimentally by us (*Mbnl1* $^{+}/Mbnl2$ $^{+}$ x *Mbnl1* $^{+}/Mbnl2$ $^{+}$). Such breeders on average produce 3-4 pups/litter. Therefore, collecting age and gender matched cohorts of *Mbnl1* $^{+}/Mbnl2$ $^{-}$ mice is challenging.

Our initial studies began with 4-month *Mbnl1* $^{+}/Mbnl2$ $^{-}$ mice as MRI studies demonstrate completion of mouse brain development at ~3 months age (31). It was subsequent to the observation of a phenotype in these animals that our interest in a comparative study of *Mbnl1* $^{-}$ /*Mbnl1* $^{+}$, *Mbnl1* $^{-}$, *Mbnl2* $^{-}$ mice was provoked. In the ~7-month period of this revision, we obtained only two female and one male *Mbnl1* $^{+}/Mbnl2$ $^{-}$ mouse from 20 breeding pairs. Therefore, identifying 2-month cohorts of five, age and gender matched *Mbnl1* $^{+}/Mbnl2$ $^{-}$ mice for this study would require breeding times that are beyond the scope of this revision. We do however agree that this comparison is valuable and we will complete this study in the future.

3. there were 3 blinded experimenters to analyze the ROI volumes (including discussions to find consensus, which is ok), but only 3 animals per group were measured (in the 2-month-old cohort). Thus, the statistical power of the results is very low. Also for the 4-month-old cohort the measured number of animals (4 control animals, i.e., the animal group that is probably the most abundant

one, and 5 animals for the experimental group) is still low. Based on this, any comparisons (conclusions) between 2-month-old and 4-month-old animals (basically between *Mbnl1*^{+/ΔE2}/*Mbnl2*^{ΔE2/ΔE2} and *Mbnl1*^{ΔE2/ΔE2}/*Mbnl2*^{+/ΔE2} are problematic.

Response-3: We agree with the Reviewer and have now added animals to our study such that all groups (both 4 and 2-months) have 5 mice/genotype. We have calculated the percent change we would be able to detect with 5 animals per group (n=10) based on a one-sample test for proportions. Assuming that, under the null hypothesis, the percent change is 1%, we would be able to detect a change of 14.4% or higher with 80% power at a Type I error rate of 5%. Similarly, for the 2-month old mice, we will be making pairwise comparisons between selected groups and would therefore be able to detect a similar percent change between any two groups. The relatively small standard deviations between subjects in some cases however allowed us to pick up significant differences ranging from ~6 to ~50% in these cohorts, which are similar to effect sizes observed in DM1 patients (3,4,22-28 & Kassubek, J *et al.* Quantification of brain atrophy in patients with myotonic dystrophy and proximal myotonic myopathy: a controlled 3-dimensional magnetic resonance imaging study. *Neuroscience Letters* **348**, 73–76 (2003), Antonini, G. *et al.* Cerebral atrophy in myotonic dystrophy: a voxel based morphometric study. *J Neurol Neurosurg Psychiatry* **75**, 1611–1613 (2004)]. In the discussion we are careful to state that direct comparisons cannot be made between the 2- and 4-month groups. We mainly focus the discussion on comparisons within the 4-month and 2-month groups.

4. One main effect was the observed change in the volume of the corpus callosum/external capsule; thus, I would have expected at least a corresponding DTI measurement (visualizing changes in the structural connectivity) and/or a histological staining to address e.g., the myelination of the axons in order to substantiate this finding. Ideally these measurements should be performed at different time points early (as early as possible after birth) and late (up to 4 month) during the development. This would at least give some hints about the role of Mbnl1/2 for postnatal development (this in turn could then also serve as start point for a subsequent study using conditioned deletion of the proteins at later time points (e.g., after 2 or 4 month).

Response-4: We agree with the Reviewer that these are interesting studies. Given that both double knockouts are rare it will take us a significant amount of time to collect the age and gender

matched cohorts required to study white matter defects at different time points with n=5/genotype/time points. As the collection of animals for this study, even with substantial numbers of breeders, is predicted to take us ~12-18 months, it is technically not feasible to complete these studies, as part of this submission.

Minor-points

5. *The used voxel size is 54.7 x 54.7 x 500 μm. These “spaghetti”-like voxels are not optimal to prevent volume effects, slice thickness of 300 μm or less would be more appropriate. An increase in averaging (from 4 – to 8 or 10) and by it an increase in total measuring time should be still tolerable for the mice.*

Response-5: The reviewer is certainly correct that higher spatial resolution and signal-to-noise ratio would be beneficial. However, as patients with DM1 can exhibit heightened sensitivity to sedatives, and anesthetics [https://www.orpha.net/data/patho/Ans/en/Myotonic-dystrophies_EN.pdf, Go, R. Wang, D. & Ludwin, D. Anesthetic Considerations in a Patient with Myotonic Dystrophy for Hip Labral Repair. Case Rep Anesthesiol 2017, 6408956 (2017)] we deemed that it would be prudent not to increase the time under anesthesia or the dose of anesthetics used in these animals. Indeed, even with the current anesthesia protocol we had two fatalities in the KO mice.

6. *Figures 3-5 and 7, y-axes often start with high values (and not at 0), thus these graphs exaggerate the effects. Furthermore, the same values/results are also presented in Table 1 and 2 (thus, Figure 5-7 are redundant to Table 1, 2 and should be moved to the supplemental part).*

Response-6: Please note that in DM1 patient MRI studies, the y-axis often starts at higher values as the effects being measured are relatively small [Please see Kassubek, J *et al.* Quantification of brain atrophy in patients with myotonic dystrophy and proximal myotonic myopathy: a controlled 3-dimensional magnetic resonance imaging study. *Neuroscience Letters* **348**, 73–76 (2003)]. The tables have all the details on a single site which will facilitate reading. As we are allowed 10 Figures/Tables, if possible, we would like to retain the Tables with the main text.

7. *Please indicate the body weights of all individual groups, are there differences between individual groups?*

Response-7: The body weights of all of the animals utilized in this study were not recorded. However, we have for other studies weighed these genotypes. In our colony *Mbn1*^{-/-}, *Mbn2*^{-/-} and *Mbn1*^{+/-}/*Mbn2*^{-/-} mice tend to obesity and weigh in general 10-15% more than the *Mbn1*^{+/+}/*Mbn2*^{+/+} controls. No overt body size differences are observed between *Mbn1*^{+/+}/*Mbn2*^{+/+} and *Mbn1*^{-/-}/*Mbn2*^{+/-} mice. *Mbn1*^{-/-}/*Mbn2*^{+/-} mice do however show prominent myotonia and appear to die due to sudden cardiac death at ~2 months of age. It should be noted that not all regions of the brain are smaller in the knockout animals. Rather they show region specific volume alterations when compared to *Mbn1*^{+/+}/*Mbn2*^{+/+} mice.

8: The abstract states: "Correlative changes in brain region integrity with behavioral alterations reported in Mbn1-/- and Mbn2-/- mice support the hypothesis that structural defects combine with altered Mbn1/2 target RNA biology to impact behavior". This is somehow misleading because in the current ms Mbn1-/- as well as Mbn2-/- mice only show minor effects on hippocampal volume (which were only evident when an additional statistical analysis was performed) and the ms presents no behavioral data. This statement should only appear in the Results and Discussion part.

Response-8: We have moved this statement to the Discussion as advised.

9. What is the difference between the here used Mbn2 ΔE2/ΔE2 mice and the Mbn2ΔE2/ΔE2 knockout mouse used by Charizanis et al doi.org/10.1016/j.neuron.2012.05.029?

Response-9: The genetic mutation is identical and Cre-mediated recombination results in both cases in the deletion of *Mbn2* exon-2. Our mice are on a 129sv background and those of Charizanis and colleagues are on a mixed 129sv/B6 background.

REVIEWERS' COMMENTS:

Reviewer #2 (Remarks to the Author):

The authors have really improved the quality of the manuscript, especially by increasing the number of animals in each group to allow meaningful comparisons. Although the number in each group is still low, I understand the problem of increasing it even more. Also, the comparison between the different groups is now better described, which makes the manuscript more readable overall. The interpretation of the data collected is comprehensible, so I have no further concerns. Just, partly out of curiosity, how long was the total measurement time when the isoflurane anesthesia was already an almost life threatening procedure.

Frank Angenstein
(DZNE site Magdeburg, Germany)

We thank the Reviewer for the evaluation of our manuscript. Response to the Reviewer is shown below:

REVIEWERS' COMMENTS:

Reviewer #2 (Remarks to the Author):

The authors have really improved the quality of the manuscript, especially by increasing the number of animals in each group to allow meaningful comparisons. Although the number in each group is still low, I understand the problem of increasing it even more. Also, the comparison between the different groups is now better described, which makes the manuscript more readable overall. The interpretation of the data collected is comprehensible, so I have no further concerns. Just, partly out of curiosity, how long was the total measurement time when the isoflurane anesthesia was already an almost life threatening procedure.

Response: Total measurement time was 30 minutes.